# Strain-driven growth of ultra-long two-dimensional nano-channels

Chao Zhu [1,2,8], Maolin Yu [3,8], Jiadong Zhou [1,8], Yongmin He [1], Qingsheng Zeng [1], Ya Deng [1], Shasha Guo [1], Mingquan Xu [2], Jinan Shi [2], Wu Zhou [2], Litao Sun [4], Lin Wang [5], Zhili Hu [3], Zhuhua Zhang [3]*, Wanlin Guo [3]* & Zheng Liu [1,6,7]*

Lateral heterostructures of two-dimensional transition metal dichalcogenides (TMDs) have offered great opportunities in the engineering of monolayer electronics, catalysis and optoelectronics. To explore the full potential of these materials, developing methods to precisely control the spatial scale of the heterostructure region is crucial. Here, we report the synthesis of ultra-long $MoS_2$ nano-channels with several micrometer length and 2–30 nanometer width within the $MoSe_2$ monolayers, based on intrinsic grain boundaries (GBs). First-principles calculations disclose that the strain fields near the GBs not only lead to the preferred substitution of selenium by sulfur but also drive coherent extension of the $MoS_2$ channel from the GBs. Such a strain-driven synthesis mechanism is further shown applicable to other topological defects. We also demonstrate that the spontaneous strain of $MoS_2$ nano-channels can further improve the hydrogen production activity of GBs, paving the way for designing GB based high-efficient TMDs in the catalytic application.

[1] School of Materials Science and Engineering, Nanyang Technological University, Singapore 639798, Singapore. [2] School of Physical Sciences, CAS Key Laboratory of Vacuum Physics, University of Chinese Academy of Sciences, Beijing 100049, China. [3] State Key Laboratory of Mechanics and Control of Mechanical Structures, Key Laboratory for Intelligent Nano Materials and Devices of Ministry of Education, and Institute of Nanoscience, Nanjing University of Aeronautics and Astronautics, Nanjing 210016, China. [4] SEU-FEI Nano-Pico Center, Key Laboratory of MEMS of Ministry of Education, Collaborative Innovation Center for Micro/Nano Fabrication, Device and System, Southeast University, Nanjing 210096, People's Republic of China. [5] Key Laboratory of Flexible Electronics & Institute of Advanced Materials, Jiangsu National Synergetic Innovation Center for Advanced Material, Nanjing Tech University, 30 South Puzhu Road, Nanjing 211816, China. [6] Environmental Chemistry and Materials Centre, Nanyang Environment and Water Research Institute, Singapore, Singapore. [7] CINTRA CNRS/NTU/THALES, UMI 3288, Research Techno Plaza, 50 Nanyang Drive, Border X Block, Level 6, Singapore 637553, Singapore. [8] These authors contributed equally: Chao Zhu, Maolin Yu, Jiadong Zhou. *email: chuwazhang@nuaa.edu.cn; wlguo@nuaa.edu.cn; z.liu@ntu.edu.sg

Transition metal dichalcogenides (TMDs) lateral hetero-structures have shown promising applications in modern semiconductor devices due to their unique electronic and optical properties[1–6] because their components are spatially separated and the band offset is tailorable[7]. However, achieving this goal requires precise control of growth not only in lattice matching but also in material dimension, considering that the Schottky barrier height, band offset, and bandgap depend vitally on the component interfacial structure and domain size[8,9]. The domain size should be within tens of nanometers or even several nanometers, which is essential for band adjustment and quantum confinement. By far, epitaxial growth has been now most widely adapted to design various lateral heterostructures, including $MoS_2$–$MoSe_2$[10], $MoS_2$–$WS_2$[11,12], $MoSe_2$–$WSe_2$[13,14], and etc. Although atomically sharp heterointerface has been demon-strated[15,16], the fabrication of nanoscale lateral heterostructures is still a great challenging because of the fast growth rate of TMDs[17–19] and the difficulties of quantitative control of vapor sources. Therefore, other growth strategies are needed to create narrow heterostructures.

Defects, mainly dislocations and grain boundaries (GBs), exist extensively in two-dimensional crystals. The previous under-standings of defects focus mostly on the analysis of atom arrangement and investigation of various properties[20–24]. Until recently, dislocations at heterointerface are found to act as cata-lysts to guide the growth of one-dimensional (1D) TMD nano-channels, making point defects a character for the growth of TMD in-plane heterostructure[25]. Unfortunately, the length of these nano-channels (<100 nm) is incomparable with the matrix flakes (usually several to tens of microns), which may restrict their practical applications. In contrast to dislocations, the microscale length and atomic-level width of 1D defects make GBs become more potential frameworks for the possible growth of flake-sized narrow heterostructure domains.

Here, using intrinsic 60° 1D GBs as the catalyst, we report the growth of ultra-long $MoS_2$ nano-channels embedded in $MoSe_2$ monolayer matrix, as shown in Fig. 1. The $MoS_2$ channels, pos-sessing atomically sharp heterointerface with $MoSe_2$ host, can reach to several micrometers long while keeping a flexible width from 2 to 30 nm. The theoretically proposed strain-driven growth mechanism based on density functional theory (DFT) calcula-tions perfectly fits into the experimental observations and maps of strain distribution.

## Results

**$MoS_2$ channel embedded in $MoSe_2$ monolayer.** Figure 2a pre-sents an optical image of the $MoSe_2$–$MoS_2$ flake, which appears to be a six-point star (light purple contrast). The formation mechanism of the star shape is due to the collision of three growing $MoSe_2$ domains with a concurrently growing larger tri-angle, as illustrated by phase-field simulation shown in Fig. 2c and Supplementary Movie 1. The simulation suggests that six GBs are starting from the star center to its six bottom concave points. Fast Fourier transformation (Fig. 2d) of the heterointer-face region shows the separated $MoS_2$ and $MoSe_2$ (110) spots with d-spacing of 0.164 and 0.158 nm, respectively. The inset in Fig. 2d shows the corresponding annular dark-field scanning transmis-sion electron microscope (ADF-STEM) image of the $MoSe_2$–$MoS_2$ heterostructure. A typical straight $MoS_2$ channel is shown in Fig. 2e. It is ~20 nm in width and at least 600 nm in length because we cannot see the entire channel due to the lim-itation of 1.2 µm holes on the gold supported foil. The channel has lower brightness than $MoSe_2$ host matrix on account of the atomic number contrast for ADF-STEM images. Such straight channels usually start at the vertexes of obtuse angles between two corners and extend toward the inside direction, as can be seen in the scanning electron microscopy (SEM) image of Fig. 2b and Supplementary Fig. 1, indicating the ultra-long feature of chan-nels (up to tens of micrometers depending on the scale of matrix flakes). High-resolution ADF-STEM image illustrates that the straight channel grows along the zigzag direction of the hexagon lattice and possess atomically sharp sidewalls coherently bonding to $MoSe_2$ monolayer (Fig. 2f, g).

**Atomic structure analysis of $MoS_2$ channel.** Further investiga-tion of the atomic structure of the rectangular area in Fig. 3a

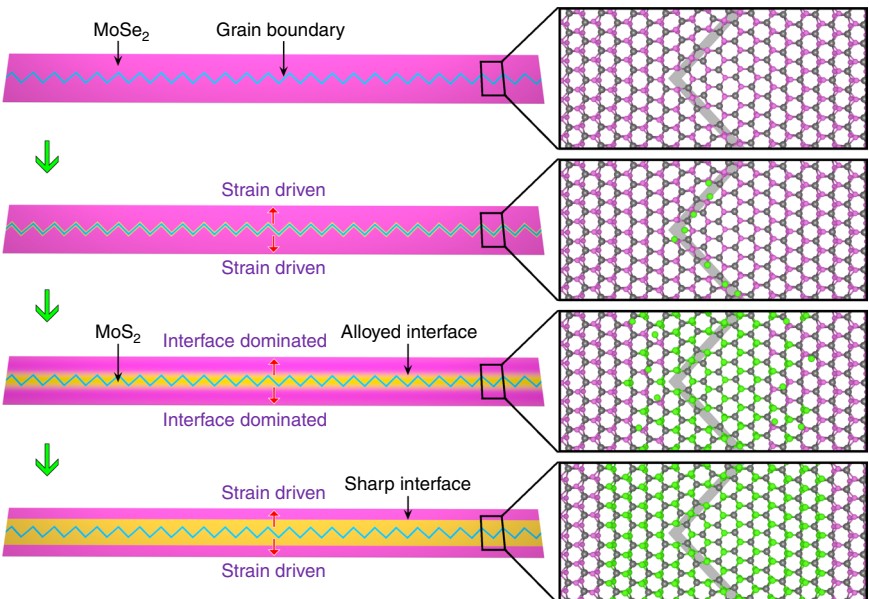

**Fig. 1 Formation of ultra-long $MoS_2$ nano-channels.** Schematic of growth steps of $MoS_2$ channels: (I) $MoSe_2$ monolayer with intrinsic 4|8 GBs is produced; (II) sulfur atoms are adsorbed and then nucleation happens due to the strain at GBs; (III) Se atoms are continuously substituted by S atoms from GBs to nearby area to form alloyed regions, and afterward the growth is dominated by interface energy; and (IV) $MoS_2$ channels with sharp interface form, and eventually the growth is sustained by the strain again to widen the channels.

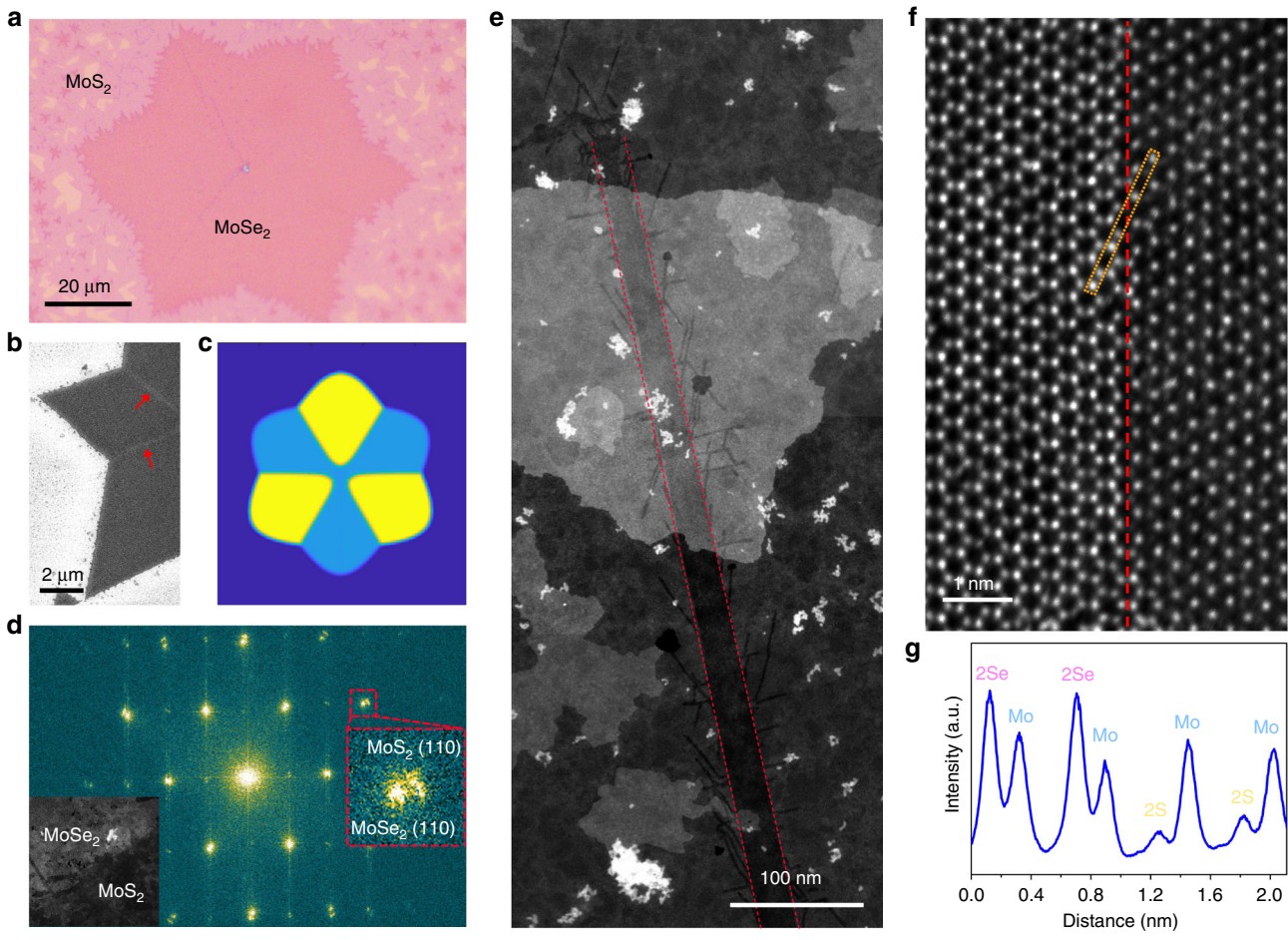

**Fig. 2 Straight MoS₂ channels embedded in MoSe₂ matrix. a** Optical image of the MoS₂–MoSe₂ lateral heterojunction, where monolayer MoS₂ surrounds the six-piont star MoSe₂. **b** SEM image showing the location of nano-channels (noted by the arrows). **c** Phase-field simulation for the growth of six-point star MoSe₂. **d** Fast Fourier transformation of the heterojunction region of the inserted ADF-STEM image. **e** Low-magnification ADF-STEM image showing a straight MoS₂ channel of >600 nm inside single-layer MoSe₂. Some brighter patches are out-of-plane MoSe₂ introduced during the transfer process. **f** ADF-STEM image of interface between MoS₂ channel and MoSe₂ matrix. The red dashed line denotes their straight and sharp interface along the zigzag direction. **g** Line intensity profile of the rectangular region in **f**.

shows that the opposite sides of the straight channel have different lattice orientations. At each sidewall (left: Fig. 3f; right: Fig. 3h), S atoms in MoS₂ channel seamlessly connect Mo atoms in MoSe₂ matrix, forming a coherent lateral interface structure. Nevertheless, a 60° (or mirror) lattice orientation difference between the opposite sides is observed, as demonstrated by the highlighted atoms in Fig. 3f, h. Careful examination of ADF-STEM images reveals that this orientation difference can be ascribed to the 60° GB in the middle region of the channel. The structure of the GB is labeled in Fig. 3g, which consists of successive fourfold and eightfold (4|8) rings. In our observation, the separated grains are mainly jointed in three ways (Supplementary Fig. 2): (I) S atoms from one grain bond Mo atoms from the other; (II) the grains share the same fourfold coordinative S atoms; (III) the grains meet at twofold coordinative S atoms. Bonds type I are found in the paired fourfold and eightfold rings, while bonds type II exist between neighboring fourfold rings, consistent with theoretical predictions and other experimental findings in TMDs[21,26]. However, bonds type III that locate between neghbouring parallel eightfold rings is seldom reported. The ring configuration is directly related to the arrangement of each type of bonds, and according to the statistical results, the amount ratio between eightfold and fourfold rings is roughly 1:1.16 (Supplementary Fig. 3). Such a ratio that is higher than

previous observation[26] should be owing to some short chains of parallel eightfold rings (Supplementary Fig. 3b).

Then, we apply the geometric phase analysis (GPA) to draw the strain fields and rotation map (see Methods section and Supplementary Fig. 4) of the channel and surrounding matrix, as plotted in Fig. 3b–e. All the phase images are obtained with reference to the MoSe₂ lattice. MoS₂ channel presents compressive $\varepsilon_{xx}$ strain (normal strain along x-direction) compared with the surrounding MoSe₂ matrix, despite that a tensile strained line is also found at the GB (Fig. 3b). For the majority area of MoS₂ channel, the calculated compressive strain is in the range of 3.9 ± 1.1%, which corresponds to the 4.2% MoS₂–MoSe₂ lattice mismatch, indicating a relaxed MoS₂ lattice along the x-direction. In contrast, MoS₂ channel and MoSe₂ host show a uniform $\varepsilon_{yy}$ strain (normal strain along y-direction) phase except the GB region (Fig. 3c), suggesting MoS₂ lattice is stretched to accommodate that of MoSe₂ in the y-direction. The shear strain ($\varepsilon_{xy}$) and lattice rotation are concentrated along the GB, as shown in Fig. 3d, e. From the rotation map, it can be clearly recognized that the GB possesses a zigzag extending trend, where the direction changing of GB leads to the altering of local orientations. Another obvious feature of the straight channel is that there are some branched MoS₂ quantum wells at the heterojunction sidewalls (Figs. 2e and 3a). These 2-nm-wide

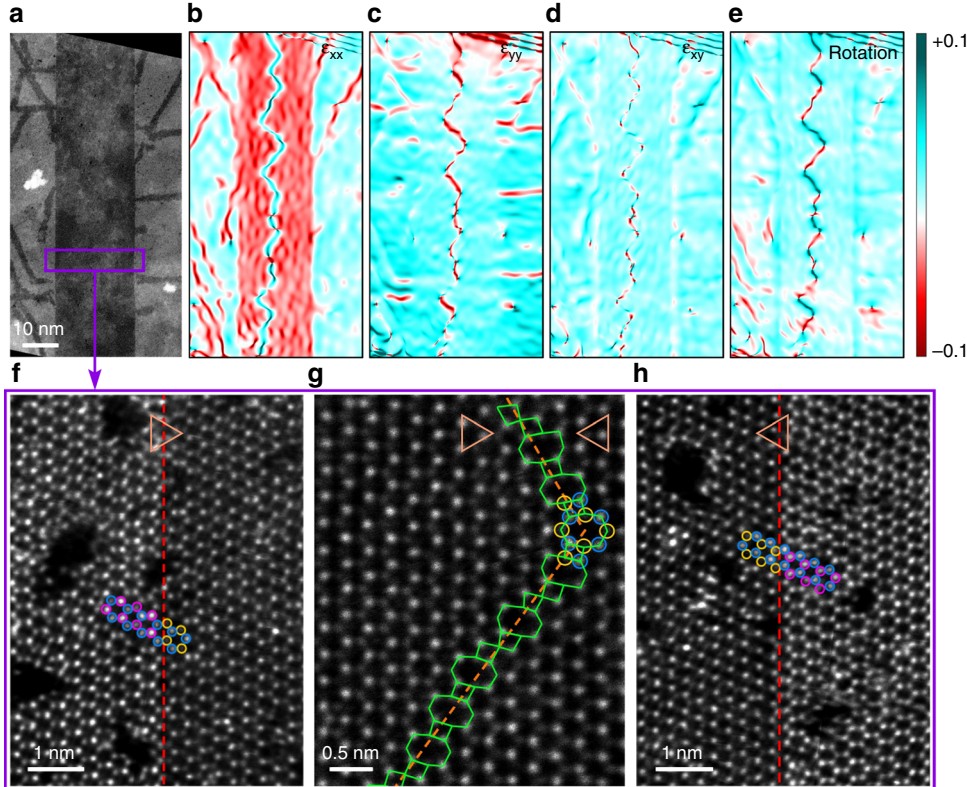

**Fig. 3 Strain maps and structure analysis of straight nano-channels. a** Low-magnification ADF-STEM image showing a part of straight MoS$_2$ nano-channel with the width of ~20 nm. **b–e** The corresponding strain (**b–d**) and rotation (**e**) maps of the channel area, showing the lattice strain and local orientations with respect to reference lattice (MoSe$_2$). A zigzag GB can be found at the middle of the straight channel. **f–h** Atomically resolved ADF-STEM images of the rectangular region in **a**. Red dashed lines denote the left (**f**) and right (**h**) interface of the channel, respectively. Orange dashed line in **g** marks the 60° GB within the channel, and it consists of alternate 4|8 rings as indicated by green tetragons and octagons. The lattice orientations at two sides of GB can be recognized from the triangles on the top and the highlighted atoms by circles (Mo, blue; Se, purple; S, yellow).

quantum wells grow along an armchair direction and always have 30° or 90° angle with the channel (Supplementary Fig. 5). The single dislocation (5|7 rings) -driven growth of such quantum wells has been investigated in recent literature[25,27]. It should be noticed that MoS$_2$ nano-channels without branched quantum wells can also be obtained through the adjustment of experimental parameters (Supplementary Fig. 6), indicating the irrelevance of these two structures.

Besides the straight channels, winding ones are also observed. A winding channel usually has a non-uniform width ranging from 2 to 20 nm (Fig. 4a). Figure 4b shows a superimposed image of the winding channel (enlarged area in Fig. 4a) and its rotation map, which complies with the extending trend of GB. By scanning the whole GB, we find that the GB segments showing the 107 ± 3° angle tend to be a periodic zigzag shape with no change of the overall GB direction, while those with 79° and 141° angles serve as a kink that turn the overall direction of the GB. To further illustrate this point, we construct a model for the GB segment with a 107° angle in Supplementary Fig. 7, where mixed 4|8 and 8|4|4|8 dislocations are neatly assembled along the GB. Such a GB is actually a consequence of seamless coalescence of two domains with mirror symmetry and has notably higher stability than those with other atomic organizations. For example, a comparison of such a folded GB with one composed fully of 8|4| 4|8 dislocations (97°) shows lower formation energy of 20–77 meV Å$^{-1}$ in the whole range of the chemical potential of sulfur. The preferred atomic organization around the GB segments folded with a 107° angle leads to their prevalence in our experimental observations. Aa a result, the winding channel has a

step-like interface with surrounding MoSe$_2$ (Fig. 4c), probably because its extending direction is not along the zigzag direction of lattice for most cases, which may also lead to the width variation of winding channel. Except for this difference, the atomic structure of the winding channel (Fig. 4c) resembles that of the straight one (Fig. 3f–h). In addition, winding and straight channels can connect with each other (Fig. 4d, Supplementary Fig. 8), suggest the similar nature of these two kinds of channels.

**Strain-driven growth mechanism based on GB.** To gain an insight into the growing mechanism, the atomic substitution process near the GBs is carefully studied by first-principles calculations. For simplicity, we consider an individual S atom for substitution reaction. In a real situation, the S-containing species may be more complicated, but the reaction is essentially the replacement of Se atoms with S atoms. Generally, the substitution of Se with S includes three steps, i.e., the chemisorption of sulfur on the MoSe$_2$ surface, an intermediate state with S and Se atoms being about to swap with each other, and the detachment of Se atoms, as illustrated by the insets in Fig. 5a. Overall, the substitution reaction is thermodynamically favorable given the higher energy of step-one configuration than the step-three one. Among the three steps, the second step is found to have the highest energy, indicating that it is the rate-limiting step throughout the whole reaction. Our proof calculations based on the energy barrier are in the same trend as that based on $\Delta E$, that is a higher $\Delta E$ correspond to a higher energy barrier presented in Supplementary Fig. 9. Apparently, a reaction site that can

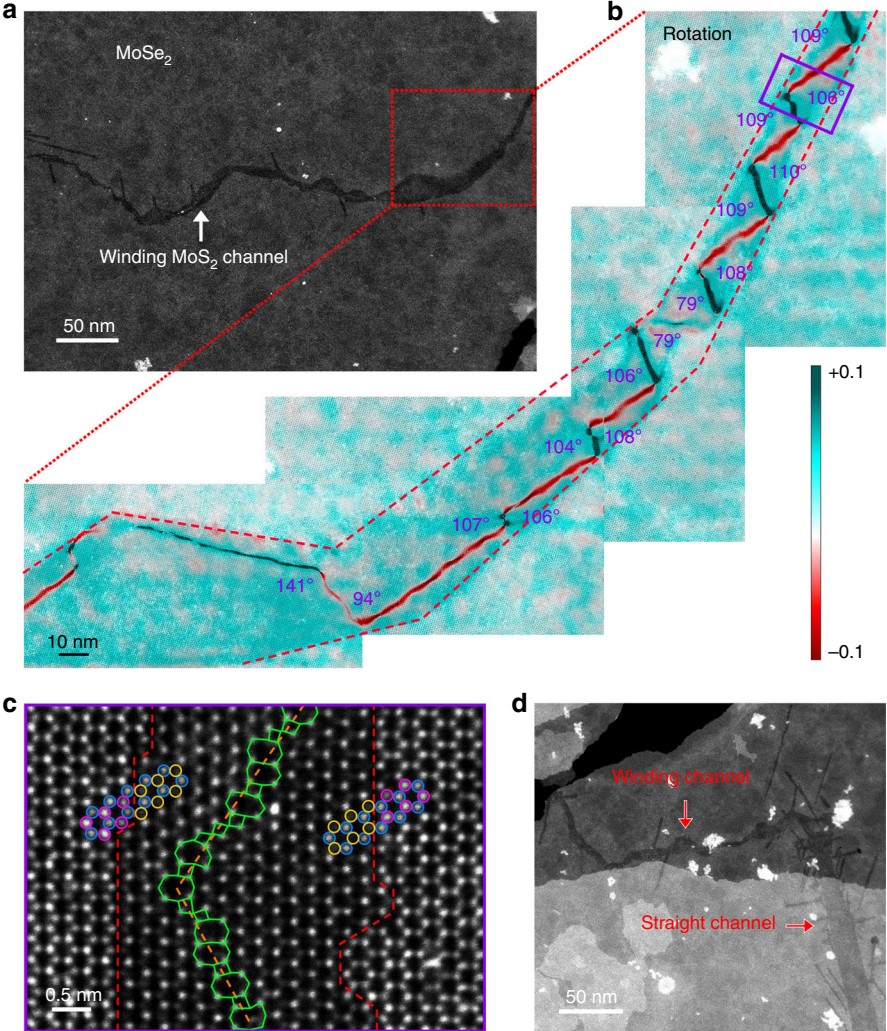

**Fig. 4 Rotation map and structure analysis of winding MoS$_2$ channels. a** Low-magnification ADF-STEM image of a winding MoS$_2$ channels. **b** The superimposed image of the channel and its corresponding rotation map. **c** High-resolution ADF-STEM image showing the atomic structure of the rectangular area in **b**. The heterojunction interface is highlighted by the red dashed lines. 60°GB (orange dashed line) is formed by 4|8 rings (green tetragons and octagons). Blue, purple, and yellow circles represent Mo, Se, and S atoms. **d** ADF-STEM image of a winding channel connecting with a straight channel.

utmostly stabilize the intermediate state could ultimately favor the reaction relative to other sites.

With this provision, we can use the relative energy of the intermediate state, $\Delta E$ (with respect to the most stable reaction site), as a reference to examine the nucleation selectivity of the MoS$_2$ channels and their dynamical evolution via the sulfur substitution. For this purpose, we first consider a GB model composed of 8|4|4|8 dislocations, the most frequently observed GB type in our experiments. A careful scan of all possible reaction sites near the GB shows that the Se atom shared by two squares will be the first to be substituted by S atom, marked by 1 in Fig. 5b. In particular, $\Delta E$, exhibits a sharp minimum of up to 1.8 eV deep as the reaction site moves toward the GB from the perfect lattice region (Fig. 5c). The strong propensity of the substitution reaction at the GB is attributed to the high in-plane tensile strain[28,29] therein (see the strain map in Fig. 3b and Supplementary Fig. 10), which greatly lowers the energy of the intermediate state due to the alleviated steric effect, while leaving the step-one configuration less influenced. Then, the S atom in MoSe$_2$ will serve as the nucleation center for the continuous

growth of the MoS$_2$. Indeed, the subsequent substitutions are energetically preferred at the Se site adjacent to the substituted S atom (marked by 2 and 3 in Fig. 5b). With a continuous supply of S atoms, the substitution of Se will not only strictly follow the GB but also extend toward the perfect lattice region. This trend is well illustrated by a contour plot of the sequence of S substitution in the MoSe$_2$ layer as shown in Fig. 5d, based on extensive calculations. This plot is essentially a translated version of the strain map around the GB. Such a strain-driven reaction mechanism well explains the MoS$_2$ channel growth in our experiments.

The growing mechanism discussed above is supported by STEM investigations at different growth steps of the GBs-based channels. Figure 6a, b displays a pristine MoSe$_2$ monolayer before S substitution (Methods section), where the same structured 60° GB composed of 4|8 rings is recognized. In addition, these GBs are located between two neighboring corners of six-point star MoSe$_2$, identical to the locations of channels in Fig. 2b. Both verify that the 60° GBs are intrinsic defects in pristine MoSe$_2$ rather than introduced structures during the growth of MoS$_2$

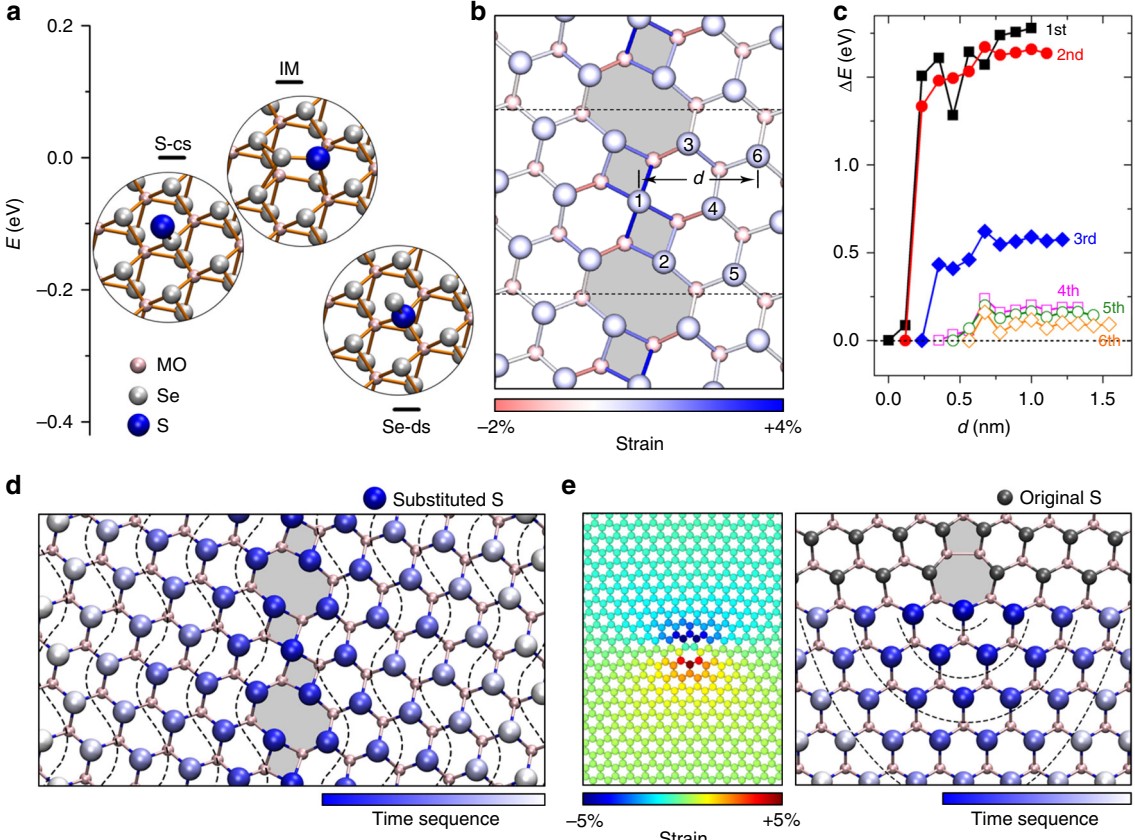

**Fig. 5 Growth mechanism of the MoS₂ channels. a** Reaction steps for the selenium substitution with sulfur and their relative energies. **b** Atomic structure of an 8|4|4|8 GB along with map of bond strain induced by the GB. The numbers 1–6 denote the sequence of sulfur substitution. $d$ denotes the distance of reaction sites with respect to the GB line. **c** Calculated relative energies $\Delta E$ for the intermediate state in **a** as a function of $d$ for the 1–6 sulfur atoms along the optimal substitution pathway. **d** Contour plot of substitution sequence (from blue to light gray) around the 8|4|4|8 GB. **e** In-plane strain distribution near a misfit Mo-rich 5|7 dislocation and the corresponding contour plot for the sequence of substitution reaction.

channels. After supplying S for a short time at ~700 °C, some Se atoms near the GB are substituted by S atoms to form a MoSe₂–MoS₂ hybrid channel (Fig. 6c). Although MoSe₂ has completely transformed into MoS₂ near the core area of the GB, there are some alloyed structures near the interfaces. The intensity profile (insert in Fig. 6d and Supplementary Fig. 11) confirms that one Mo atom randomly coordinates with S₂, Se₂, or S + Se atoms in the alloyed regions, as labeled by the arrows in Fig. 6d. Yet, no alloyed structure is observed in the region away from the interfaces, implying that GBs are the nucleation sites for the channel growth. Moreover, some chemisorptive S atoms can be identified at the MoSe₂–MoS₂ interface (Supplementary Fig. 12). These observations agree with our theoretical scenario that the S substitution starts at the GBs and then spreads toward the bulk regions, guided by the long-range strain fields pertaining to the GBs.

The strain-driven reaction mechanism can be applied to other sources of lattice strain, such as dislocations and point defects. Different lattice constants of MoS₂ and MoSe₂ result in a misfit Mo-rich 5|7 dislocation at their interface, which can induce an even higher long-range strain field than that by GBs, as evidenced by the strain map in the left panel of Fig. 5e. The calculated substitution sequence according to energy criteria reveals a strong preference of reaction at the dislocations and a substitution pathway closely following the strain map (Supplementary Fig. 13). As a result, the growth of MoS₂ channels at the interface branch off from the dislocations, forming a unique sub-channel spreading into the MoSe₂ areas, as observed experimentally in

Fig. 2e. Besides the dislocations, we also purposefully create Mo and Se vacancies in the MoSe₂, which induce strain only in the immediate vicinity of the defects. As expected, the growth of MoS₂ does occur near the defects but spreads only by several atoms away from the vacancies (Supplementary Fig. 14). We envision that the bias of chemical reaction can also be amenable to applied elastic strain, which potentially enables larger flexibility for fabricating MoS₂ superstructures with a wide variety of patterns.

Furthermore, it is worth mentioning that the growth of MoS₂ channels ends up with a sharp heterointerface after long-time reaction. This can be understood by the minimized interface energy. After the growth of channels advances far from the GB, the effect of GB-induced strain is diminished (Supplementary Fig. 15), and MoSe₂–MoS₂ interfaces gradually dominate the extension of the channels. As such, the growth will proceed in a manner that minimizes the interface energy, thereby resulting in a sharp, straight interface. Firstly, our DFT calculations prove that a zigzag-shaped MoSe₂–MoS₂ heterointerface will be gradually smoothened upon the substitution reaction and finally evolve into a straight line, as shown in Fig. 6e. Then, the strain always plays a major role in sustaining the growth of the MoS₂ channel. At the sharp interface between MoS₂ and MoSe₂ in Supplementary Fig. 16, scanning all possible reaction sites shows that the Se atom closest to the interface will be the first to be substituted by S atom, followed by the Se atoms on the same row, with $\Delta E$ exhibiting a sharp minimum of 0.05–0.15 eV deep. The overall growth process of MoS₂ channels are schematically illustrated in Fig. 1, vividly

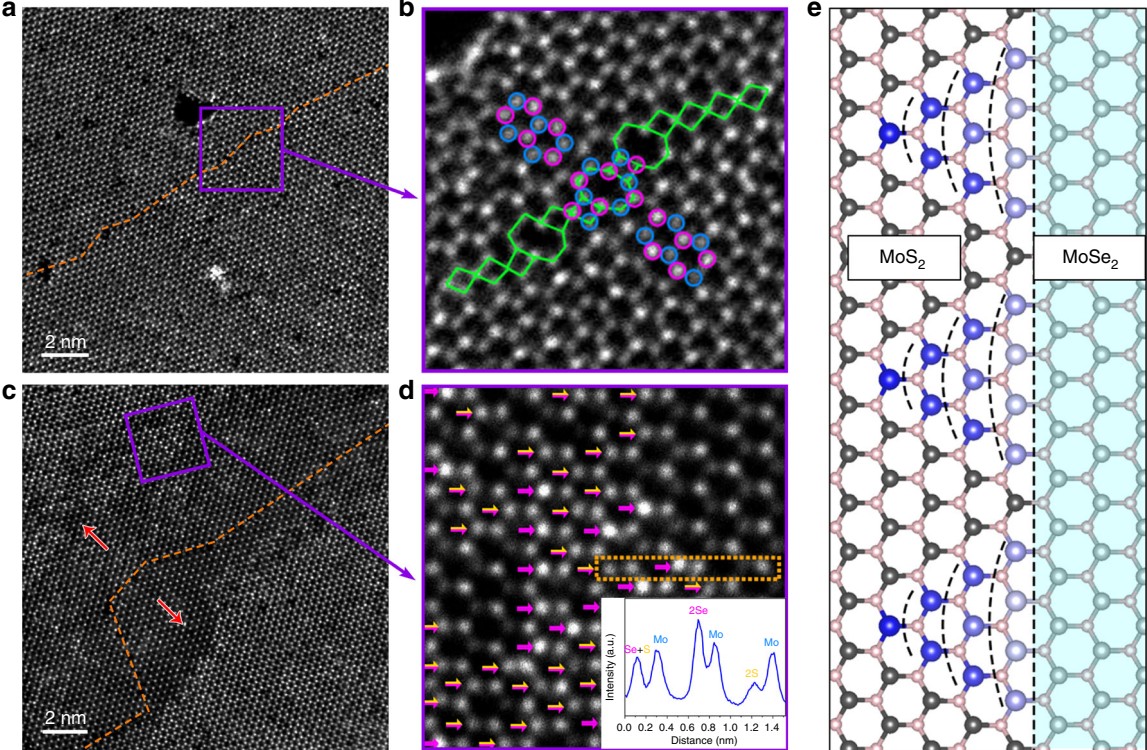

**Fig. 6 Growth process of MoS₂ channels. a, b** ADF-STEM images of intrinsic 60° GB (orange dashed line) within pristine monolayer MoSe₂. Atomic arrangement of this GB (blue and purple circles: Mo and Se atoms; and green tetragons and octagons: 4 and 8 rings) resembles that in MoS₂ nano-channels. **c, d** ADF-STEM images showing part of MoSe₂–MoS₂ hybrid winding channel. Some Se atoms has been replaced by S atoms, as indicated in **d** (purple arrows mark 2Se atoms and yellow–purple arrows mark S + Se atoms). Line intensity profile of 2Se, 2S, S + Se, and Mo atoms is provided in the insert of **d**. The red arrows in **c** represent the sulfidation direction. **e** A contour plot for the sequence (from blue to light gray) of substitution reaction along a zigzag MoSe₂–MoS₂ interface.

displaying a transition from the GB-driven growth mechanism to an interface-dominated mechanism.

**Catalytic performance of nano-channels for HER.** It has been reported that GBs in TMDs possess better catalytic performance in hydrogen evolution reaction (HER) than intact lattice[30]. In addition, recent works also prove the connection between the catalytic activity and the strain of crystal lattice for kinds of TMDs[31,32]. In the consideration of these facts, it is reasonable to speculate that the combination of strain and GBs should be an effective strategy for further improvement of the activity in hydrogen production. To our delight, the MoS₂ nano-channels are appropriate structures that applying a y-direction 3.9% lattice-mismatch strain along GBs (Fig. 3c). Moreover, it should be noticed that this method is a spontaneous process, only if any GB exists in MoSe₂, MoS₂ nano-channel can be produced accurately along the GB to introduce the lattice-mismatch strain, which cannot be accomplished by other external strain engineering strategies, such as stretching[33] and bending[34]. In order to investigate the activity of the spontaneous strained MoS₂ nano-channels, we develop a micro-electrochemical device to exactly examine the hydrogen production performance (Supplementary Fig. 17), as shown in Fig. 7a. For comparison, the same devices are also fabricated on common single GB and basal plane in pure MoS₂ (Supplementary Fig. 18). Figure 7b, c presents the polarization curves and the corresponding Tafel slopes in 0.5 M H₂SO₄ solution, respectively, for three different devices (Pt is also included as a reference). It can be seen that although the common GB in pure MoS₂ has achieved a better catalytic activity than the

basal plane, the spontaneous strained nano-channel even exhibits a more improved performance. The statistical data (Fig. 7d) based on tens of devices strengthen the reliability of our results, confirming that the spontaneous strain in MoS₂ nano-channels can further improve the catalytic activity of GBs.

## Discussion

In conclusion, we have demonstrated the growth of ultra-long nano-channels based on GBs, and the length of the nano-channels is comparable with the matrix. We proposed a strain-driven growing mechanism to elucidates the role of defects (both point and line defects) in the formation of the nano-channels. The nano-channels have also been proved to introduce spontaneous strain at GBs for efficient improvement of catalytic performance in hydrogen production, and also for bandgap altering (Supplementary Fig. 19). Our work has suggested a different strategy to realize narrow heterostructures with great potential for bandgap engineering, quantum effect, and electrocatalysis such as the formation of 1D charge density waves[35], gate-tunable memristive phenomena[36], and strong photoluminescence quenching/enhancement[26].

## Methods

**Synthesis procedure.** The MoS₂–MoSe₂ lateral heterostructures were synthesized by a two-step chemical vapor deposition (CVD) method using MoO₃, Se, and S powders as the Mo, Se, and S source, respectively. The first step: the growth of MoSe₂ flakes. Specifically, a clean SiO₂/Si substrate was placed face-down onto a quartz boat containing 15 mg MoO₃ powders. The boat was loaded into the middle of a quartz tube with an inner diameter of 1 inch. Another quartz boat containing 0.5 g Se powder was placed upstream. The growth was performed at atmospheric pressure with a mixture of 5 sccm H₂ and 50 sccm Ar as the carrier gas. The

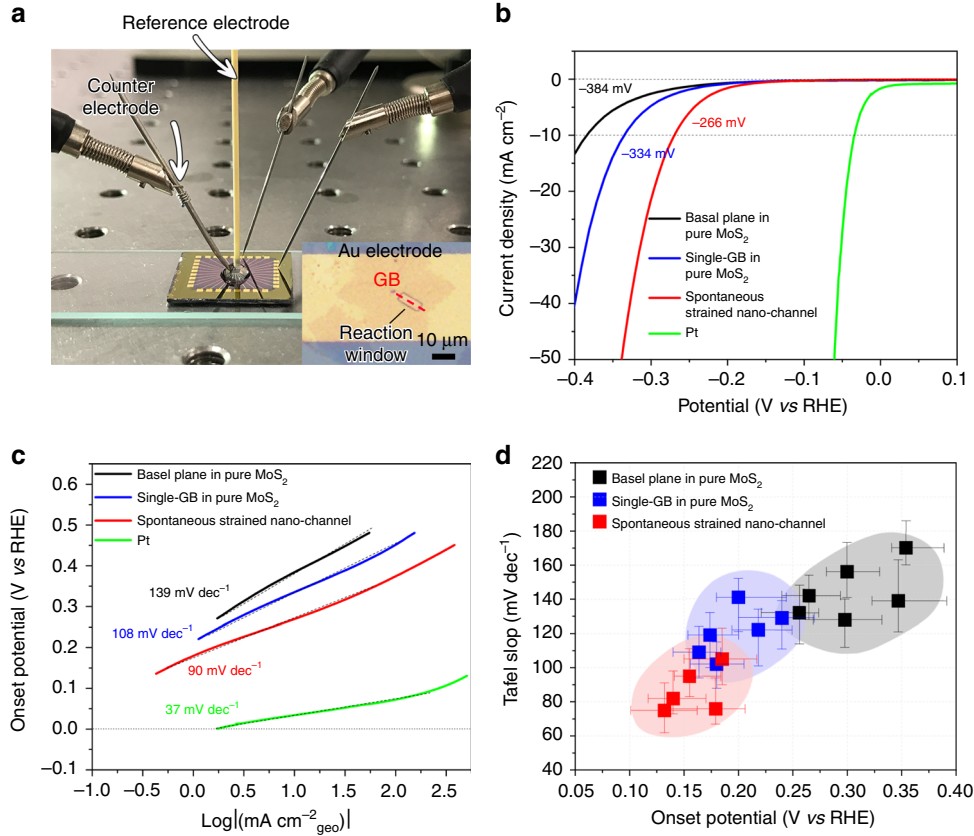

**Fig. 7 HER activity of MoS₂ single GB in nano-channel. a** Photograph of micro-electrochemical cell. Inset: device with a micro-size reaction window at the spontaneous strained nano-channel. The hydrogen evolution reaction (HER) only occurs within the reaction window, and the rest of the areas are passivated by electrochemically inert PMMA film. **b–c** Polarization curves of the current density (**b**) and the corresponding Tafel plots (**c**) of the devices for spontaneous strained MoS₂ nano-channel, single GB, and basal plane in pure MoS₂, respectively. **d** Statistical HER results obtained from dozens of micro-electrochemical devices. The error bars define the data statistical range of multiple measurements for each micro-electrochemical cell.

temperature ramped up to 720 °C at a rate of 30 °C min⁻¹, and the duration time at 720 °C was set to 10 min. After that, the furnace was cooled down to room temperature naturally. The second step: the growth of heterostructures. Specifically, the cooled substrate was immediately transferred to another quartz tube for MoS₂ monolayer growth. During this step, 15 mg MoO₃ and 0.5 g S powder were used as the precursors and the gas flow kept the same as in the MoS₂ case. The growth was carried out at 700 °C for 5 min.

**STEM sample preparation.** The STEM samples were transferred using a poly (methyl methacrylate) (PMMA) assisted method. A thin layer of 1 μm PMMA was spin coated onto the SiO₂/Si substrate and then baked at 120 °C for 3 min. The substrate was immersed in 20% KOH solution to etch the SiO₂ layer, resulting in a floating PMMA/MoS₂–MoSe₂ film. After washing with deionized water, the film was fished by a Au TEM grid (Quantifoil, 50 nm Au foil of 300 mesh). Finally, PMMA coating layer was dissolved by soaking the grid in 60 °C acetone for 3 h. Before STEM characterization, the grid was annealed under Ar atmosphere at 200 °C for 3 h to avoid carbon hydrocarbon contamination.

**STEM characterization.** STEM characterization was carried out on a JEOL ARM-200F (S)TEM equipped with CEOS CESCOR aberration corrector, operated at an accelerating voltage of 80 kV. The convergence semi-angle and acquisition semi-angle were 28–33 and 68–280 mrad for the ADF imaging. The dwell time per pixel was set to 12–20 μs. The atomic resolution ADF images were deconvolution filtered using Richard-Lucy method to enhance the contrast.

**Geometric phase analysis.** GPA is a standard method to quantitatively extract displacement fields and strain maps from high-resolution TEM images. This method is based on the Fourier transformation and inverse Fourier transformation because the lattice information of real space is described by the corresponding peaks in the reciprocal space. Here, we applied GPA to map the strain fields of MoS₂ nano-channels embedded in MoSe₂ matrix. Firstly, two strong nonparallel Bragg reflections $g_1$ and $g_2$ in FFT images are selected and covered by Gaussian masks (Supplementary Fig. 4b). The followed calculation was performed to

generate the phase images (Supplementary Fig. 4c, d) and scale of reciprocal lattice (Supplementary Fig. 4e, f) corresponding to reflections $g_1$ and $g_2$, as well as the strain and rotation maps (Supplementary Fig. 4g–j).

**Theoretical calculations.** All the theoretical calculations were performed with the Vienna Ab-initio Simulation Package[37]. The projector-augmented wave method for the core region and the generalized gradient approximation with the Perdew, Burke, and Ernzerhof functional were employed. Kinetic energy cutoff of 300 eV was adopted in the plane-wave expansion. The model systems are constructed in nanoribbon configurations, in which the distance between GBs and edges is ∼7 nm, large enough to achieve the convergence of the energies. All structures are fully relaxed until the force on each atom is <0.01 eV Å⁻¹. The Brillouin zone integration was sampled by five special $k$-points along the periodic orientation for the 8|4|4|8 GBs and gamma-only points for the 5|7 dislocations.

The phase-field setups are almost identical to that in ref. [38]. We digest the adaptions we adopted for the current simulation. The model[39,40] starts from a free energy functional,

$$G = \sum_{\alpha,\beta=1,2,\alpha<\beta} \frac{4\sigma_{\alpha\beta}}{\pi^2}\left(-\eta\nabla\phi_\alpha \times \nabla\phi_\beta + \frac{\pi^2}{\eta}\phi_\alpha\phi_\beta\right) + \lambda\xi h(\phi_N) \qquad (1)$$

Variables description can be found in the publication[38]. In the current simulation, we set $N = 3$ thus two grain orientations are considered. The simulation early stage is virtually the growth of a David star[38] on a 256 × 256 canvas. Each nucleus starts from a circle with a radius of 3$l$. Distances between central nuclei and satellite nuclei are 10$l$. Once the length of David star reaches 85% of edge length, the flux is set to zero so the growth ceases and etching starts. The simulation stops when the length of David star drops to 30% of edge length. The parameters are listed in Supplementary Table 1.

**Fabrication of micro-electrochemical device and electrocatalytic measurement.** First, a prepatterned set of 32 Au contact pads was fabricated on a 16 mm × 16 mm SiO₂ (285 nm)/Si chip using conventional photolithography (Supplementary Fig. 17a). Second, MoS₂/MoS₂–MoSe₂ film from the CVD growth method was

transferred onto this chip (Supplementary Fig. 17b), and a further annealing process at 200 °C under high-vacuum conditions ($1 \times 10^{-5}$ torr) was employed to optimize their interfaces to facilitate the electron vertical injection from Au to the sample during reaction (Supplementary Fig. 17c). Third, a 1-μm-thick PMMA film was coated on the device chip, and an e-beam lithography process was followed to open a reaction window on this film to expose the region of interest on the nanosheet for reaction (Supplementary Fig. 17d). The location of single GB can be clearly identified by Raman mapping or dark-field optical microscope.

A micro-electrochemical device with four electrodes was adopted in the electrocatalytic experiment (Fig. 7a). In all measurements, only the exposed region of the sample contributes to HER performance. The measurements were conducted in a 0.5 M $H_2SO_4$ electrolyte solution. The scan rate was set to be 5 mv per step. The electrocatalytic current ($I_c$) and conductance current ($I_{ds}$) are simultaneously detected.

## Data availability
The data that support the plots within this paper and other findings of this study are available from the corresponding authors upon reasonable request

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

## Acknowledgements
This work is supported by the Singapore National Research Foundation under NRF award number NRF-RF2013-08, MOE Tier 2 MOE2016-T2-2-153, MOE2015-T2-2-007, MOE2015-T2-2-043, and MOE2018-T3-1-002. The part of work in NUAA is supported by NSF of China (51535005, 51472117, and 11772153), the Research Fund of State Key Laboratory of Mechanics and Control of Mechanical Structures (MCMS-I-0418K01, MCMS-0417G02, and MCMS-0417G01). The part of work in UCAS is supported by NSF of China (51622211 and 51872285).

## Author contributions
C.Z. and Z.L. conceived the idea. C.Z. performed the electron microscopy experiments, electron microscopy data analysis, and wrote the paper. M.Y., Z.Z., Z.H., and W.G. performed the theoretical calculations and participated in the paper writing. J.Z. synthesized the MoSe₂–MS₂ heterostructure samples. Y.H., Y.D., and S.G. carried out the electrochemical experiment. W.Z., C.Z., M.X., and J.S. performed the electron energy loss spectroscopy (EELS) experiment. Q.Z., L.W., and L.S. participate in the data analysis. All authors discussed the results and commented on the manuscript.

## Competing interests
The authors declare no competing interests.
