## [Peer Review File · Nature Communications]

Reviewers' comments:

Reviewer #1 (Remarks to the Author):

The authors studied the grain boundaries in lateral heterostructures of TMDs synthesized using chemical vapor deposition (CVD) method. A detailed investigation on the boundaries using STEM and DFT calculation was performed. The authors proposed a new concept: nano-channels based on the intrinsic grain boundaries. The formation mechanism of the channel is attributed to the strain in the lattice close to the boundaries. But still, I find the nature of the work is lack of novelty. The synthesis of 2D TMDs and lateral heterostructures has been widely reported. The quality of the materials presented in this work does not look good, suggesting lots of defects in the materials. I am very concerned about the reproducibility of different channels in this work, thus the potential of these "channels" may not be so useful as the authors claimed. The authors should demonstrate an application of the channel in this work to strength the work to enable to publish on Nature Communications. In addition, although the STEM investigations on the grain boundaries are pretty systematic, similar investigations were also reported in many other literature. Therefore, I would suggest to publish this work on other journals, such as ACS Nano and Nano Letters. Below are some questions I have:

1. In most of the TEM/STEM images, the formation of the MoS₂ quantum wells was observed. With these quantum well branches on the side, the edge of the as-synthesized nano-channel is not clearly defined, especially for the narrow nano-channels. Since the formation of the quantum well is due to the dislocation at the MoS₂-MoSe₂ interface, how will the growth of the quantum well relate to the nano-channel growth? Are there factors that may affect the quantum well distribution along the nano-channel?
2. In the discussion of the formation of winding channels, the authors mentioned that grain boundaries with 107 deg. between the segments tend to grow straight MoS₂ channel. Is there any structural reason of this specific angle to be the criteria of forming a straight channel? What is the relation between '107 deg.' and the structure or geometry of the grain boundaries?

Reviewer #2 (Remarks to the Author):

The manuscript reports the growth of relatively straight MoS₂ ribbons within MoSe₂ monolayers via S atom substitution initiated at grain boundaries. The energetic favorability of the proposed pathway is investigated with density functional theory calculations. Analysis of differences in the strain of bonds in the vicinity of the defects supports the claim that the growth is "strain driven".

Thinner sub-nm MoS₂ channels are also observed as was reported previously in references 25 and 27; reference 25 also reported MoS₂ formation at a grain boundary. The primary distinction in the present work is widening of initially rough boundaries into surprisingly straight sidewalls, which minimizes the interfacial energy.

Overall, there is sufficient novelty in this result to warrant publication, even if the level of control is likely to be poor for the powder based CVD process. The authors should address a few points in revision.

1. Though the microscopic analysis is convincing regarding where and why growth initiates, it might be too simplistic to say that the the growth of overall MoS₂ ribbon is strain driven unless the authors can explain the final width in terms of reduction in the magnitude of the driving force upon widening. The abrupt interface that forms in the final structure may not minimize strain- one would expect a graded interface to minimize strain energy. It is important to note that the exchange is also driven, thermodynamically, by an excess of S.

2. There is a very high density of defects, particularly voids, in the surrounding material. Presumably there are vacancies and other point defects as well. What role might vacancies play in the kinetics of growth/exchange?
3. Is it possible to grow the ribbons without introducing nanochannels driven by dislocations?
4. The standard term is "geometric phase analysis" rather than "geometry phase analysis"

Reviewer #3 (Remarks to the Author):

This manuscript reports the growth of ultra-long MoS₂ nano-channels embedded in MoSe₂ host matrix using the intrinsic grain boundaries as catalyst. The straight and atomically sharp interface between MoS₂ channel and the hosting MoSe₂ was unveiled by the SDF-STEM techniques. I believe that this is an important piece of work in the growing and promising applications of TMDs and would recommend the publication of this work on Nature Communications upon addressing the following issues:

- 1)The authors explain that the synthesis mechanism of MoS₂ nano-channels is strain-driven. The intrinsic GBs consisting of 4|8 rings can indeed induce a strain field as evidenced by the strain field maps. But it seems that the atomic substitution process occurring near the GBs is more relevant to the chemical reactivity of the imperfect lattices than the strain at the GBs.
- 2)Since the electronic and optical properties of the in-plane heterostructures are attractive, would the authors do the Raman or photoluminescence characterizations to reveal the optical properties of MoS₂ nano-channels embedded in MoSe₂?
- 3)The energy barrier of intermediate state can be calculated by the vasp software using the neb or ci-neb methods. For a reaction site, the energy barrier for substitution of Se with S atoms should be one and only. I didn't understand how the authors unravel the nucleation selectivity of MoS₂ by calculating ΔE as a function of d .
- 4)In Fig. 4c, there are the brighter atoms in the red dashed line and they have the similar contrast as the Se atoms. Why the authors denote them to S atoms?
- 5)The MoS₂ nano-channel is formed after S substitution around the GBs. So, how to prove all the Se atoms are replaced with S atoms, not just the Se atoms on the top layer of the sandwich TMD structure.

Response to the reviewers' comments

Reviewer #1 (Remarks to the Author):

The authors studied the grain boundaries in lateral heterostructures of TMDs synthesized using chemical vapor deposition (CVD) method. A detailed investigation on the boundaries using STEM and DFT calculation was performed. The authors proposed a new concept: nano-channels based on the intrinsic grain boundaries. The formation mechanism of the channel is attributed to the strain in the lattice close to the boundaries. But still, I find the nature of the work is lack of novelty. The synthesis of 2D TMDs and lateral heterostructures has been widely reported. The quality of the materials presented in this work does not look good, suggesting lots of defects in the materials. I am very concerned about the reproducibility of different channels in this work, thus the potential of these “channels” may not be so useful as the authors claimed. The authors should demonstrate an application of the channel in this work to strength the work to enable to publish on Nature Communications. In addition, although the STEM investigations on the grain boundaries are pretty systematic, similar investigations were also reported in many other literature. Therefore, I would suggest to publish this work on other journals, such as ACS Nano and Nano Letters. Below are some questions I have:

Authors' Reply

We thank the reviewer for this valuable suggestion. The reviewer raised three questions about (1) the work is lack of novelty. The synthesis of 2D TMDs and lateral heterojunctions has been widely reported, and the STEM investigation on GBs were reported in many other literature, (2) the quality of the materials and reproducibility of different channels and (3) demonstrate an application of the channel to strength the work. We reply them separately below:

(1) The work is lack of novelty. The synthesis of 2D TMDs and lateral heterojunctions has been widely reported, and the STEM investigation on GBs were reported in many other literature.

Although lateral heterojunctions and the investigation of GBs have been reported in many other literature, our unique GB induced nano-channels have never been reported. The novelty of this work relies on the following points:

(i) The novelty of structure and growth mechanism. On one hand, almost all the reported lateral heterojunctions, including MoS₂-MoSe₂, MoS₂-WS₂, MoSe₂-WSe₂, and etc (Table R1.1), are synthesized through epitaxial method: the growth of one monolayer always starts from the edges of another monolayer.

However, our proposed mechanism is totally different from the previous ones: the growth of one monolayer starts from the GBs of another monolayer and then driven by strain field, finally lead to the forming of hetero-nano-channels. We not only explain how the nano-channels grow, but also elucidate the evolution of heterointerface from roughness to sharpness, which is a major distinction from other works, agreed by the Review #3's statement **“The primary distinction in the present work is widening of initially rough boundaries into surprisingly straight sidewalls, which minimizes the interfacial energy. Overall, there is sufficient novelty in this result to warrant publication.”** On the other hand, most of the reported STEM investigation of GBs focus on the atomic structure analysis of them, but our focus is the function of GBs during heterojunction growth, including the nucleation site and strain effect. Therefore, our nano-channel structure and strain-driven mechanism are unique and have never been reported yet, making the work possess enough originality and novelty.

Types of heterostructure	Growing mechanism	
	Epitaxial growth	Strain-driven growth
MoS ₂ -MoSe ₂	Reference 1	This work
WSe ₂ -MoSe ₂	Reference 2-4	
MoS ₂ -WS ₂	Reference 4-6	
WSe ₂ -MoS ₂	Reference 7	
WS ₂ -WSe ₂	Reference 8, 9	
ReSe ₂ -ReS ₂	Reference 10	

Table R1.1. Summary of lateral heterojunctions synthesized by epitaxial and strain-driven method.

(ii) The novelty of potential application in catalysis. MoS₂ have been proved to show great potentials in electrocatalysis because all the catalytically active sites are exposed due to the atomically thin nature of the nanosheets (Fig. R1.1). In compared to intact lattice, GBs in TMDs possess even better catalytic performance¹¹, which always encourage researchers to synthesize defect-abundant monolayers for application of electrocatalysis. Besides, recent works also prove that the hydrogen adsorption free energy is highly related to the strain of crystal lattice for kinds of TMDs (MX₂, M = Nb, Ta, Mo, W; X = S, Se, Te)¹².

Fig. R1.1. The development of TMD catalysts. **a**, Volcano plot of the exchange current density as a function of the Gibbs free energy of adsorbed atomic hydrogen for nanoparticulate MoS₂ and the pure metals in 2007¹⁴. **b**, Review of TMD catalysts in 2016¹⁵.

Based on the above publications, we believe that **the combination of strain and GBs should be an effective strategy for further improvement of the activity in hydrogen production.** To our delight, the MoS₂ nano-channels are appropriate structures that applying a considerably lattice-mismatch strain at GBs. As presented in Fig. R1.2a and the manuscript description, MoS₂ lattice of nano-channel is stretched to accommodate that of MoSe₂ in the *y*-direction, but relaxed along *x*-direction, indicating a *y*-direction normal strain of nano-channels. This is also confirmed by the FFT (Fig. R1.2b), where the (110) spots of MoS₂ and MoSe₂ overlap in the *y*-direction (left enlarged spots), while separate in other direction with *x*-component (right enlarged spots). All these data suggest a 3.9% lattice-mismatch strain at GBs within MoS₂ nano-channels, making them excellent candidate for HER catalysis (please see this application in Fig. R1.4 and R1.5). **Moreover, it should be noticed that our method is a new “spontaneous” process: only if any GB exists in MoSe₂, MoS₂ nano-channel can be produced accurately along the GB to introduce the lattice-mismatch strain for the enhancement of catalytic property, that cannot be accomplished by other external strain engineering strategies, such as stretching¹⁶ and bending¹⁷**

Fig. R1.2. “Spontaneous” strain at MoS₂ GBs. **a**, Strain maps (ϵ_{xx} and ϵ_{yy}) intercepted from Fig. 3 in manuscript showing the “spontaneous” strain along the direction (*y*-direction) of nano-channels. **b**, The

corresponding FFT of **a**.

(2) The quality of the materials and reproducibility of different channels.

Indeed, a high density of vacancies and voids can be seen at some areas, but these defects are not intrinsic features of our samples. They are mainly introduced during sample transfer and treatment: (1) Before STEM image capturing, we carry out a beam shower process¹⁸ to prevent carbon contamination which is a major disadvantage for high resolution imaging of monolayer materials. This process can effectively reduce carbon deposition onto samples by applying a high electron flux, but at the same time inevitably make some damage to the samples since vacancies and vacancy complexes can be easily generated at TMD monolayers even at 60 kV¹⁹ (we operate at 80 kV). (2) During the transport process, in order to remove the organic contamination, sometimes 5% HF is used for sample cleaning. This process further increases the vulnerability of monolayers. So generally, most of the vacancies and voids (Fig. 3f.h) are introduced by electron irradiation. Being aware of this, we have tried our best to reduce the damage from these sources through decreasing the duration of beam shower and HF free transfer. As a result, nearly intact monolayers can be obtained, shown in Fig. R1.3 (dark points at MoSe₂ lattice are dopant S atoms, rather than vacancies).

The growth of quantum wells and nano-channels has a good reproducibility. We can now synthesize pure quantum wells and nano-channels in a controlled way, as shown in Fig. R1.6 and R1.7.

Fig. R1.3. Large field of view ADF-STEM images of nano-channels with low density of vacancies and voids.

(3) Demonstrate an application of the channel to strength the work.

Here, we demonstrate that the “spontaneous” strain of MoS₂ nano-channels can further improve the

hydrogen-production activity of MoS₂ GBs, which already have excellent catalytic performance. In order to investigate the activity of the strained GBs in the nano-channels, we develop a micro-electrochemical device to exactly examine the hydrogen-production performance of single GB in a nano-channel, as shown in Fig. R1.4a. For comparison, the same devices are also fabricated on common single-GB and basal plane in pure MoS₂ (Fig. R1.5). Fig. R1.4b, c presents the polarization curves and the corresponding Tafel slopes in 0.5 M H₂SO₄ solution respectively, for three different devices (Pt is also included as a reference). **It can be seen that although the common GB in pure MoS₂ has achieved good catalytic activity than the basal plane, the “spontaneous” strained nano-channel can further improve the performance.** The statistical data (Fig. R1.4d) based on tens of devices strengthen the reliability of our results, confirming that the “spontaneous” strain in MoS₂ nano-channels makes GBs high catalytic activity. This finding suggests that our method could be used for “spontaneous” strain engineering of GBs, paving the way of GB based high-efficient catalyst for hydrogen production. We have added this part in revised manuscript (Fig. 7) and supplementary information.

Fig. R1.4. HER activity of MoS₂ single-GB in nano-channel. a, Photograph of micro-electrochemical cell. Inset: device with a micro-size reaction window at the “spontaneous” strained nano-channel. The hydrogen evolution reaction (HER) only occurs within the reaction window, and the rest of the areas are

passivated by electrochemically-inert PMMA film. **b-c**, Polarization curves of the current density (**b**) and the corresponding Tafel plots (**c**) of the devices for “spontaneous” strained MoS₂ nano-channel, single-GB and basal plane in pure MoS₂ respectively. **d**, Statistical HER results obtained from dozens of micro-electrochemical devices.

Fig. R1.5. Optical images of pure MoS₂ micro-electrochemical devices. The reaction windows are opened on single-GB (**a**) and basal plane (**b**) for HER measurement.

Comment 1: In most of the TEM/STEM images, the formation of the MoS₂ quantum wells was observed. With these quantum well branches on the side, the edge of the as-synthesized nano-channel is not clearly defined, especially for the narrow nano-channels. Since the formation of the quantum well is due to the dislocation at the MoS₂-MoSe₂ interface, how will the growth of the quantum well relate to the nano-channel growth? Are there factors that may affect the quantum well distribution along the nano-channel?

Authors' Reply

We thank the reviewer for raising a question on the definition of “nano-channel” and “quantum well branches”. Although both nano-channels and quantum wells are produced through the driven force of strain, they are based on different defects: 1D GB for the former and 0D dislocation for the lateral. Because of the different lattice parameters of MoS₂ and MoSe₂, a lot of dislocations exist at their hetero-interface to accommodate the lattice mismatch. Here, we ignore a very small amount of isolated dislocations out of hetero-interface, as shown in Supplementary Fig. 5c-g. Hence, the prerequisite to growth dislocation-related quantum wells is the formation of MoS₂-MoSe₂ lateral hetero-structures. This can be confirmed by the ADF-STEM images (Fig. R1.6a and Supplementary Fig. 8b), where all the quantum wells grow from the hetero-interface. Similarly, when GB-induced nano-channels begin to grow, the hetero-interface will form at the same time, finally leading to the growth of branched quantum wells on the side. Therefore, growth of nano-channels precedes the growth of branched quantum wells, and these quantum wells just come out from the side of nano-channels, will not affect the formation of nano-channels.

Fig. R1.6. Solely growth of quantum wells and nano-channels. **a**, A high density of quantum wells without nano-channels. **b**, Nano-channels without quantum wells. Dashed red lines denote the hetero-interface between MoS₂ and MoSe₂.

The more straightforward evidence is that we can synthesize pure quantum wells and nano-channels in a controlled way, as shown in Fig. R1.6. We found that the growth of quantum wells is much more sensitive to temperature: high temperature leads to more and longer quantum well; low temperature leads to less and shorter quantum wells. Whereas, temperature has little influence on nano-channel growth. Finally, high-density quantum wells can be obtained at ~780 °C (Fig. R1.6a), and nano-channels without branched quantum wells can be obtained at ~670 °C (Fig. R1.6b), which evidence they are two different phenomena that can be well controlled.

The detailed structure characterization of a MoS₂ nano-channel without branched quantum wells is shown in Fig. R1.7. A nano-channel (highlighted by the arrows) with the width of 4 nm and length of more than 1 μm can be seen in Fig. R1.7a. Not any branched quantum well could be found along this channel (Fig. R1.7b), and a zigzag GB resides in it (Fig. R1.7c). Careful investigation reveals the 4|8 ring constructed GB and different orientation of the lattice at two sides (Fig. R1.7d). The FFT indicates a 55° GB instead of 60° GB (Fig. R1.7e), further suggesting the universality of our proposed strain-driven mechanism. We have added this discussion in revised manuscript and supplementary information.

Fig. R1.7. MoS₂ nano-channel without quantum wells obtained at high temperature. a, Large field of view ADF-STEM image of a nano-channel without quantum wells. **b**, Enlarged ADF-STEM image of large rectangular region in **a**. **c**, Enlarged ADF-STEM image and corresponding strain map of small rectangular region in **a**. **d**, High resolution ADF-STEM image of this nano-channel showing its atomic structure. **e**, Fast Fourier transformation of **c**.

Comment 2: In the discussion of the formation of winding channels, the authors mentioned that grain boundaries with 107 deg. between the segments tend to grow straight MoS₂ channel. Is there any structural reason of this specific angle to be the criteria of forming a straight channel? What is the relation between ‘107 deg.’ and the structure or geometry of the grain boundaries?

Authors’ Reply

Thanks for the comment. By scanning the whole grain boundary (GB), we find that the GB segments showing the 107° angle tend to be a periodic zigzag shape with no change of the overall GB direction, while those with 79° and 141° angles serve as a kink that turn the overall direction of the GB. To further illustrate this point, we construct a model for the GB segment with a 107° angle in Fig. R1.8, where mixed 4|8 and 8|4|4|8 dislocations are neatly assembled along the GB. Such a GB is actually a consequence of seamless coalescence of two domains with a mirror symmetry and has notably higher stability than those

with other atomic organizations. For example, a comparison of such a folded GB with one composed fully of 8|4|4|8 dislocations (97°) shows a lower formation energy of 20 - 77 meV/Å in the whole range of the chemical potential of sulfur. Therefore, the preferred atomic organization around the GB segments folded with a 107° angle leads to their prevalence in our experimental observations. We have discussed this in the revised manuscript.

Fig. R1.8. Atomic structures of folded GBs. **a**, The one composed of solely of the 8|4|4|8 dislocations. **b**, The GB composed of mixed 8|4 + 8|4|4|8 dislocations. **c**, Energies of 97° GB relative to 107° GB as functions of the chemical potential of sulfur in the range $-5.4 \text{ eV} < \mu_s < -4.1 \text{ eV}$.

Reviewer #2 (Remarks to the Author):

The manuscript reports the growth of relatively straight MoS2 ribbons within MoSe2 monolayers via S atom substitution initiated at grain boundaries. The energetic favorability of the proposed pathway is investigated with density functional theory calculations. Analysis of differences in the strain of bonds in the vicinity of the defects supports the claim that the growth is "strain driven". Thinner sub-nm MoS2 channels are also observed as was reported previously in references 25 and 27; reference 25 also reported MoS2 formation at a grain boundary. The primary distinction in the present work is widening of initially rough boundaries into surprisingly straight sidewalls, which minimizes the interfacial energy. Overall, there is sufficient novelty in this result to warrant publication, even if the level of control is likely to be poor for the powder based CVD process. The authors should address a few points in revision.

Authors' Reply

We highly appreciate the Reviewer's positive comments on our work.

Comment 1: Though the microscopic analysis is convincing regarding where and why growth initiates, it might be too simplistic to say that the growth of overall MoS₂ ribbon is strain driven unless the authors can explain the final width in terms of reduction in the magnitude of the driving force upon widening. The abrupt interface that forms in the final structure may not minimize strain- one would expect a graded interface to minimize strain energy. It is important to note that the exchange is also driven, thermodynamically, by an excess of S.

Authors' Reply

Thanks for the insightful comment. We would like to point out that the strain always plays a major role in regulating the growth of the MoS₂ ribbon, via two stages. In the first stage, the growth is initiated by the high strain field around the GB. Therefore, this driving force diminishes gradually as the growth frontier moves away from the GB (see the energy change in Fig. 5c in the manuscript). The second stage would be the growth from the sharp interface. To clarify this point, we perform first-principles calculations for the atomic exchange at the sharp interface between MoS₂ and MoSe₂, with results shown in Fig. R2.1. Scanning all possible reaction sites shows that the Se atom closest to the interface will be the first to be substituted by S atom, marked by 1 in Fig. R2.1a. This is because the interfacial Se atoms are connected by strained chemical bonds due to the abrupt chemical change. Then, the inclusion of the first S introduces extra tensile strain, which benefits the substitution of 2nd Se atoms in the same row. In particular, ΔE exhibits a sharp minimum of 0.05-0.15 eV deep at the interface for all the examined substitutions (Fig. R2.1b). Therefore, the driving force for the growth of a MoS₂ channel includes two regimes dominated by the GB and interface, respectively, both are correlated with lattice strain. We have added this discussion in revised manuscript and supplementary information.

Fig. R2.1 a, Atomic structure of the perfect MoS₂/MoSe₂ interface. The numbers 1~3 denote the sequence

of S substitution. d denotes the distance of reaction sites with respect to the interface line. **b**, Calculated relative energies ΔE for the intermediate state in **a** as a function of d for the 1~3 S atoms along the optimal substitution pathway.

Comment 2: There is a very high density of defects, particularly voids, in the surrounding material. Presumably there are vacancies and other point defects as well. What role might vacancies play in the kinetics of growth/exchange?

Authors' Reply

We thank the reviewer for raising this question. Indeed, a high density of vacancies and voids can be seen at some areas, but these defects are not intrinsic features of our samples. They are mainly introduced during sample transfer and treatment: (1) Before STEM image capturing, we carry out a beam shower process¹⁸ to prevent carbon contamination which is a major disadvantage for high resolution imaging of monolayer materials. This process can effectively reduce carbon deposition onto samples by applying a high electron flux, but at the same time inevitably make some damage to the samples since vacancies and vacancy complexes can be easily generated at TMD monolayers even at 60 kV¹⁹ (we operate at 80 kV). (2) During the transport process, in order to remove the organic contamination, sometimes 5% HF is used for sample cleaning. This process further increases the vulnerability of monolayers. So generally, most of the vacancies and voids (Fig. 3f,h) are introduced by electron irradiation. Being aware of this, we have tried our best to reduce the damage from these sources through decreasing the duration of beam shower and HF free transfer. As a result, nearly intact monolayers can be obtained, shown in Fig. R2.2a (dark points at MoSe₂ lattice are dopant S atoms, rather than vacancies).

In addition, even if these are some defects, the growth of nano-channels will not be affected. In our controlled experiment, we make a large amount of vacancies and voids by plasma treatment of MoSe₂, followed by growth of MoS₂. Apparent MoS₂ channels based on GBs (Fig. R2.2c) can still be produced, highlighted by the arrows in Fig. R2.2b. All the manufactured defects become MoS₂ islands (Fig. R2.2d and Supplementary Fig. 13), suggesting that vacancies and voids are “repaired” during MoS₂ growth. Therefore, although vacancies can introduce some MoS₂ islands, they are irrelevant with the growth of nano-channels.

Fig. R2.2. Vacancy influence during nano-channel growth. **a**, Large field of view ADF-STEM images of nano-channels with low density of vacancies and voids. **b**, ADF-STEM image of a MoS₂ channel based on plasma-treated MoSe₂ matrix. **c**, Rotation map of **b**. **d**, Enlarged ADF-STEM image of the rectangular location in **b**.

Comment 3: Is it possible to grow the ribbons without introducing nanochannels driven by dislocations?

Authors' Reply

We thank the reviewer for raising this question. Although both nano-channels and quantum wells are produced through the driven force of strain, they are based on different defects: 1D GB for the former and 0D dislocation for the latter. Because of the different lattice parameters of MoS₂ and MoSe₂, a lot of dislocations exist at their hetero-interface to accommodate the lattice mismatch. Here, we ignore a very small amount of isolated dislocations out of hetero-interface, as shown in Supplementary Fig. 5c-g. Hence, the prerequisite to growth dislocation-related quantum wells is the formation of MoS₂-MoSe₂ lateral hetero-structures. This can be confirmed by the ADF-STEM images (Fig. R2.3a and Supplementary Fig. 8b), where all the quantum wells grow from the hetero-interface. Similarly, when GB-induced nano-channels begin to grow, the hetero-interface will form at the same time, finally leading to the growth of branched quantum wells on the side. Therefore, growth of nano-channels precedes the growth of

branched quantum wells, and these quantum wells just come out from the side of nano-channels, will not affect the formation of nano-channels.

Fig. R2.3. Solely growth of quantum wells and nano-channels. **a**, A high density of quantum wells without nano-channels. **b**, Nano-channels without quantum wells. Dashed red lines denote the hetero-interface between MoS₂ and MoSe₂.

As a result, we can synthesize pure quantum wells and nano-channels in a controlled way, as shown in Fig. R2.3. We found that the growth of quantum wells is much more sensitive to temperature: high temperature leads to more and longer quantum well; low temperature leads to less and shorter quantum wells. Whereas, temperature has little influence on nano-channel growth. Finally, high-density quantum wells can be obtained at ~ 780 °C (Fig. R2.3a), and nano-channels without branched quantum wells can be obtained at ~ 670 °C (Fig. R2.3b), which evidence they are two different phenomena that can be well controlled.

The detailed structure characterization of a MoS₂ nano-channel without branched quantum wells is shown in Fig. R2.4. A nano-channel (highlighted by the arrows) with the width of 4 nm and length of more than 1 μm can be seen in Fig. R2.4a. Not any branched quantum well could be found along this channel (Fig. R2.4b), and a zigzag GB resides in it (Fig. R2.4c). Careful investigation reveals the 4|8 ring constructed GB and different orientation of the lattice at two sides (Fig. R2.4d). The FFT indicates a 55° GB instead of 60° GB (Fig. R2.4e), further suggesting the universality of our proposed strain-driven mechanism. We have added this discussion in revised manuscript and supplementary information.

Fig. R2.4. MoS₂ nano-channel without quantum wells obtained at high temperature. **a**, Large field of view ADF-STEM image of a nano-channel without quantum wells. **b**, Enlarged ADF-STEM image of large rectangular region in **a**. **c**, Enlarged ADF-STEM image and corresponding strain map of small rectangular region in **a**. **d**, High resolution ADF-STEM image of this nano-channel showing its atomic structure. **e**, Fast Fourier transformation of **c**.

Comment 4: The standard term is "geometric phase analysis" rather than "geometry phase analysis"

Authors' Reply

We thank the reviewer for pointing out this mistake. We have modified all the "geometry phase analysis" to "geometric phase analysis" both in manuscript and supplementary information.

Reviewer #3 (Remarks to the Author):

This manuscript reports the growth of ultra-long MoS₂ nano-channels embedded in MoSe₂ host matrix using the intrinsic grain boundaries as catalyst. The straight and atomically sharp interface between MoS₂ channel and the hosting MoSe₂ was unveiled by the SDF-STEM techniques. I believe that this is an

important piece of work in the growing and promising applications of TMDs and would recommend the publication of this work on Nature Communications upon addressing the following issues:

Authors' Reply

We highly appreciate the Reviewer's positive comments on our work.

Comment 1: The authors explain that the synthesis mechanism of MoS₂ nano-channels is strain-driven. The intrinsic GBs consisting of 4|8 rings can indeed induce a strain field as evidenced by the strain field maps. But it seems that the atomic substitution process occurring near the GBs is more relevant to the chemical reactivity of the imperfect lattices than the strain at the GBs.

Authors' Reply

It is true that the chemical reactivity near the GBs or other defects is high. However, enhanced chemical reactivity is more attributed to the lattice strain since i) all atoms near the GBs are fully coordinated without dangling bonds, and ii) the reaction remains energetically preferred at the growth frontier that has been progressed away from the GB. To further reflect the correlation between the chemical reactivity and strain, we calculated the variation of absorption energy of H atoms on a MoSe₂ sheet shown in Fig. R3.1. The absorption energy of H atoms drops by up to ~0.6 eV when a moderate 3% tensile strain is applied. In comparison, the lattice strain near the GB can be up to 4%.

Fig. R3.1 Evolution of absorption energy of H atoms on a MoSe₂ sheet against applied lattice strain. The adsorption energy is relative to the energy of an individual H atom.

Comment 2: Since the electronic and optical properties of the in-plane heterostructures are attractive, would the authors do the Raman or photoluminescence characterizations to reveal the optical properties of MoS₂ nano-channels embedded in MoSe₂?

Authors' Reply

We thank the reviewer for this suggestion. We have carried out the spatially resolved Raman mapping in WITEC alpha 200R Raman system. 532 nm excitation laser and 1800 lines/nm grating were used with an approximately 1 μm spot sizes, and the laser power was kept below 1mW to avoid the damage of samples. The Raman mapping reveals the spatial modulation within the MoS₂-MoSe₂ hetero-structure flake (A_{1g} peaks), with the center part composed of a 6-point star domain of MoSe₂ (Fig. R3.2b) and the peripheral part consisting of MoS₂ (Fig. R3.2c), corresponding to the contrast optical image (Fig. R3.2a). Six apparent nano-channels (GBs) can be clearly imaged from the MoSe₂ modulation, highlighted by the arrow in Fig. R3.2b, indicating the different optical properties between MoS₂ channels (GBs) and MoSe₂ matrix. However, no clear nano-channels can be obtained from the MoS₂ modulation (Fig. R3.2b), which may be due to the weak signal of our narrow nano-channels (several nanometers) compared with the background signal of surrounding MoSe₂.

Fig. R3.2 Optical property of MoS₂ nano-channels. **a**, Optical image of MoS₂-MoSe₂ lateral hetero-structures. **b,c**, Corresponding spatially resolved Raman mapping images show the MoS₂(**b**)-MoSe₂(**c**) hetero-structure signals.

Comment 3: The energy barrier of intermediate state can be calculated by the vasp software using the neb or ci-neb methods. For a reaction site, the energy barrier for substitution of Se with S atoms should be one and only. I didn't understand how the authors unravel the nucleation selectivity of MoS₂ by calculating ΔE as a function of d.

Authors' Reply

Yes, the nucleation selectivity of MoS₂ is determined by the energy barrier of the overall reaction process which can be calculated by VASP. However, calculating such energy barriers for all considered reaction sites (~200 sites) is time-consuming and computationally unaffordable. As mentioned in the manuscript, this energy barrier must be proportional to the energy, ΔE , of the intermediate state relative to the initial adsorption state. We therefore calculate ΔE to quantify the reaction selectivity among different sites, instead of energy barriers. To justify this approximation, we calculate the energy barriers of the 1st, 2nd and 6th sites considered for substituting the first Se atom, as shown in Fig. R3.3. The energy barrier of 1st site is 0.53 eV, followed by 0.82 eV for the 2nd one and 1.29 eV for 6th one. Thus, the results based on the energy barrier are in the same trend as that based on ΔE , that is a higher ΔE correspond to a higher energy barrier. Therefore, we can use ΔE as a reference to examine the nucleation selectivity and their dynamical evolution via the sulfur substitution while reducing the computational cost. We have added this discussion in revised manuscript and supplementary information.

Fig. R3.3 Calculated minimum energy pathways energy for taking the 1st, 2nd and 6th Se atom as the substitution sites for the first S atom (see Fig. 5b). The intermediate state is marked by IM in each energy pathway.

Comment 4: In Fig. 4c, there are the brighter atoms in the red dashed line and they have the similar contrast as the Se atoms. Why the authors denote them to S atoms?

Authors' Reply

We thank the reviewer for this comment and we have replaced Fig. 4c with a new one. Actually, by careful investigation of some winding channels, it is found that not all the Se atoms can be substituted by S atoms, leading to some doped region in MoS₂ channels (highlighted by the circles in Fig. R3.4a). Most of these doped atoms are SeS rather than SeSe atom column (highlighted by the arrows in Fig. R3.4b), confirmed

by the contrast analysis from Fig. R3.5. We speculate this may arise from: (1) the bottom Se atoms are a little bit harder to be substituted than top Se atoms because of the interaction with substrate; (2) some residual Se source absorbed onto substrate or MoSe₂, in turn, replace S atoms during MoS₂ growth process. However, despite of some impurities, the Se doping density of MoS₂ channel is very low, ~ 0.36% (calculated from a length of 50 nm MoS₂ channel), which can be ignored in the analysis.

Fig. R3.4. ADF-STEM images of MoS₂ winding channels. a, Large field of view ADF-STEM images showing some doped region in MoS₂ channels. **b**, High resolution ADF-STEM image showing the atomic structure of a winding channel.

Comment 5: The MoS₂ nano-channel is formed after S substitution around the GBs. So, how to prove all the Se atoms are replaced with S atoms, not just the Se atoms on the top layer of the sandwich TMD structure.

Authors' Reply

To address this question, we carry out the comparison between experimental and simulated ADF-STEM images. The contrast of incoherent ADF-STEM image is proportional to $Z^{1.6-1.7}$, where Z is the atomic number²⁰. Accordingly, the contrast ratio of Mo, SeS, SS and SeSe could be estimated to be 1 : 0.89 : 0.39 : 1.40. Fig. R3.5a shows the atomic model of MoSeS-MoS₂-MoSe₂ hetero-structures, where it can be seen from the side view that the top layer of MoSeS is S atoms whereas the bottom layer is Se atoms. In the

simulated image (Fig. R3.5b), the contrast of Mo, SeS, SS and SeSe (blue lines in Fig. R3.5d and e) approximately correspond to the above calculated ratio. As expected, in the experimental image (Fig. R3.5c), the contrast of Mo-SeS-Mo-2S (orange circles in Fig. R3.5d) and Mo-2S-Mo-2Se (pink circles in Fig. R3.5e) atom chain is also in line with that of the simulated image. Therefore, we can easily distinguish the SeS, SS and SeSe atom column due their apparent contrast difference, so that to confirm if all the Se atoms are replaced by S atoms. This method has been used to determine the atomic structure of some alloyed regions (Fig. 6d). We have added this image in revised supplementary information.

Fig. R3.5. Determination of SeS, SS and SeSe atom column through ADF-STEM images. **a**, Atomic model of lateral hetero-structure of monolayer MoSeS-MoS₂-MoSe₂. **b**, Simulated ADF-STEM image corresponding to the rectangular region of **a**. **c**, Experimental ADF-STEM image of some doped region in MoS₂ channel. **d**, Line intensity profile of simulated (orange rectangle in **b**) and experimental (orange rectangle in **c**) atom chains of Mo-SeS-Mo-2S. **e**, Line intensity profile of simulated (pink rectangle in **b**) and experimental (pink rectangle in **c**) atom chains of Mo-2S-Mo-2Se.

1. Duan X, *et al.* Lateral epitaxial growth of two-dimensional layered semiconductor heterojunctions. *Nature nanotechnology* **9**, 1024 (2014).
2. Gong Y, *et al.* Two-step growth of two-dimensional WSe₂/MoSe₂ heterostructures. *Nano Lett* **15**, 6135-6141 (2015).
3. Ullah F, *et al.* Growth and Simultaneous Valleys Manipulation of Two-Dimensional MoSe₂-WSe₂ Lateral Heterostructure. *ACS nano* **11**, 8822-8829 (2017).
4. Zhang X-Q, Lin C-H, Tseng Y-W, Huang K-H, Lee Y-H. Synthesis of lateral heterostructures of semiconducting atomic layers. *Nano Lett* **15**, 410-415 (2014).
5. Gong Y, *et al.* Vertical and in-plane heterostructures from WS₂/MoS₂ monolayers. *Nature materials* **13**, 1135-1142 (2014).
6. Sahoo PK, Memaran S, Xin Y, Balicas L, Gutiérrez HR. One-pot growth of two-dimensional lateral heterostructures via sequential edge-epitaxy. *Nature* **553**, 63 (2018).
7. Li M-Y, *et al.* Epitaxial growth of a monolayer WSe₂-MoS₂ lateral pn junction with an atomically sharp interface. *Science* **349**, 524-528 (2015).
8. Zhang Z, Chen P, Duan X, Zang K, Luo J, Duan X. Robust epitaxial growth of two-dimensional heterostructures, multiheterostructures, and superlattices. *Science* **357**, 788-792 (2017).
9. Xie S, *et al.* Coherent, atomically thin transition-metal dichalcogenide superlattices with engineered strain. *Science* **359**, 1131-1136 (2018).
10. Liu D, *et al.* Diverse Atomically Sharp Interfaces and Linear Dichroism of 1T'ReS₂□ReSe₂ Lateral p-n Heterojunctions. *Adv Funct Mater* **28**, 1804696 (2018).
11. Zhu J, *et al.* Boundary activated hydrogen evolution reaction on monolayer MoS₂. *Nature communications* **10**, (2019).
12. Li H, *et al.* Activating and optimizing MoS₂ basal planes for hydrogen evolution through the formation of strained sulphur vacancies. *Nature materials* **15**, 48 (2016).
13. Zhao S, *et al.* Group VB transition metal dichalcogenides for oxygen reduction reaction and strain-enhanced activity governed by p-orbital electrons of chalcogen. *Nano Research* **12**, 925-930 (2019).
14. Jaramillo TF, Jørgensen KP, Bonde J, Nielsen JH, Hørch S, Chorkendorff I. Identification of active edge sites for electrochemical H₂ evolution from MoS₂ nanocatalysts. *Science* **317**, 100-102 (2007).
15. Voiry D, Yang J, Chhowalla M. Recent Strategies for Improving the Catalytic Activity of 2D TMD Nanosheets Toward the Hydrogen Evolution Reaction. *Adv Mater* **28**, 6197-6206 (2016).
16. Castellanos-Gomez A, *et al.* Local strain engineering in atomically thin MoS₂. *Nano Lett* **13**, 5361-5366 (2013).
17. Zhang Q, *et al.* Strain relaxation of monolayer WS₂ on plastic substrate. *Adv Funct Mater* **26**, 8707-8714 (2016).
18. Mitchell DR. Contamination mitigation strategies for scanning transmission electron microscopy. *Micron* **73**, 36-46 (2015).
19. Lin J, *et al.* Flexible metallic nanowires with self-adaptive contacts to semiconducting transition-metal dichalcogenide monolayers. *Nature nanotechnology* **9**, 436-442 (2014).
20. Krivanek OL, *et al.* Atom-by-atom structural and chemical analysis by annular dark-field electron microscopy. *Nature* **464**, 571 (2010).

Reviewers' comments:

Reviewer #1 (Remarks to the Author):

The authors did a significant improvement to the manuscript and demonstrated better HER performance based on the nano-channels.

Reviewer #2 (Remarks to the Author):

The authors have done a thorough job responding to referee comments. The manuscript is suitable for publication.

Reviewer #3 (Remarks to the Author):

In their revised version, the authors have considerably improved their manuscript, and addressed most of the issues raised by the reviewers. And fabrication of micro-electrochemical device and electrocatalytic measurement have been added in the revised version.

However, the experimental proofs are still not enough clear to support their strain-driven claim. There is no experimental evidence to prove that the driving force diminishes as the growth frontier moves away from the GBs. And the Fig.5c could only prove the energy barrier of the S substitution reaction near the GBs is lower. Given that the 4|8 GBs are intrinsically chemically active, the S substitution reaction couldn't be simply attributed to the strain-driven mechanism. As to the substitution reaction, the lattice strain is more likely to be the result, rather than the cause of it.

In the caption of Fig.2c, the growth of 6-point star MoS₂ or MoSe₂?

Reviewer #4 (Remarks to the Author):

The authors report the formation of lateral heterojunction 2D TMDs by the substitution of Se with S starting from G.B.s in the matrix of MoSe₂. the manuscript is self-consistent and can be published in Nature Comm. A critical concern is about Novelty. Their method in the manuscript is not new and also not much improved in terms of quality of experimental results they've presented. The reviewer agreed with one of the reviewer's comment on Novelty. "The primary distinction in the present work is widening of initially rough boundaries into surprisingly straight sidewalls,..." In the manuscript, their analysis to support the experiemental results can be considered a scientific advancement.

For the application of lateral heterojunction as strained nanochannels, the authors choose HER and demonstrated better catalytic performance with micron sized electrochemical reactors. No objection with their experiements and results as well, but HER is not a good choice for the lateral heterojunction in 2D TMDs. It is known that surfaces and/or edge sites where dangling bonds are located show much better catalytic behaviors compared with anyother in 2D TMDs. the reviewer recomend careful electrical transportation experiments through the heterojunction are much more critical and can show some potential electronic devices applications.

Response to the reviewers' comments

Reviewer #1 (Remarks to the Author):

The authors did a significant improvement to the manuscript and demonstrated better HER performance based on the nano-channels.

Authors' Reply

We appreciate the reviewer's positive comments on our HER demonstration.

Reviewer #2 (Remarks to the Author):

The authors have done a thorough job responding to referee comments. The manuscript is suitable for publication.

Authors' Reply

We thank the reviewer for the positive comments.

Reviewer #3 (Remarks to the Author):

In their revised version, the authors have considerably improved their manuscript, and addressed most of the issues raised by the reviewers. And fabrication of micro-electrochemical device and electrocatalytic measurement have been added in the revised version.

Authors' Reply

We thank the reviewer for the positive comments on our revision.

Comment 1: However, the experimental proofs are still not enough clear to support their strain-driven claim. There is no experimental evidence to prove that the driving force diminishes as the growth frontier moves away from the GBs. And the Fig.5c could only prove the energy barrier of the S substitution

reaction near the GBs is lower. Given that the 4|8 GBs are intrinsically chemically active, the S substitution reaction couldn't be simply attributed to the strain-driven mechanism. As to the substitution reaction, the lattice strain is more likely to be the result, rather than the cause of it.

Authors' Reply

We thank the reviewer for raising the questions about (1) the experimental proofs are still not clear enough to support the strain-driven claim, (2) Fig.5c could only prove the energy barrier of the S substitution reaction near the GBs is lower. Given that the 4|8 GBs are intrinsically chemically active, the S substitution reaction couldn't be simply attributed to the strain-driven mechanism. We reply to them separately below:

(1) The experimental proofs are still not enough clear to support the strain-driven claim. There is no experimental evidence to prove that the driving force diminishes as the growth frontier moves away from the GBs.

To confirm the diminishing of the driving force as the growth frontier moves away from the GBs, we analyzed the strain of the intermediate growth state of MoS₂ nano-channels. The sample growth was terminated at the early stage and then the sample was transferred on grids for STEM characterization. Fig. R1a shows a partly-formed nano-channel and the corresponding strain maps. It can be seen that almost all the Se atoms have been substituted by S atoms near the GB, and the alloyed growth frontier (highlighted by the yellow lines) is 1 ~ 2 nm away from the GB. The line profiles obtained from the green line regions of the ϵ_{xx} and ϵ_{yy} strain maps are shown in Fig. R1b and c, respectively. At the GB, significant compressive and tensile strain have been observed in ϵ_{xx} and ϵ_{yy} maps, respectively, which leads to the occurrence of substitution reaction, in good accordance with our theoretical calculation. However, the strain decreases quickly as the distance to the GB increases. The high strain region only has a range of ~ 1.4 nm for ϵ_{xx} (Fig. R1b) and ~ 1.8 nm for ϵ_{yy} (Fig. R1c). In addition, there is no other high strain region observed near the growth frontier, implying the interface energy dominated growth at this stage as discussed in the main text (line 233-241). Therefore, it can be confirmed from both experimental and theoretical results that the driving force diminishes as the growth frontier moves away from the GB, and more specifically, the driving force can be completely ignored when the growth frontier is > 1 nm away from the GB. We have added the data and discussion in the supplementary information.

Fig. R1. Strain analysis of the growth intermediate state of MoS₂ nano-channels. **a**, ADF-STEM image of a partly-formed MoS₂ nano-channel embedded in MoSe₂, and the corresponding strain maps. The GB is noted with dashed lines and the growth frontier is highlighted with yellow solid lines. **b**, The line profile from the green line region in ϵ_{xx} strain map. **c**, The line profile from the green line region in ϵ_{yy} strain map.

(2) Fig.5c could only prove the energy barrier of the S substitution reaction near the GBs is lower. Given that the 4|8 GBs are intrinsically chemically active, the S substitution reaction couldn't be simply attributed to the strain-driven mechanism.

It is right that the S substitution reaction couldn't be simply attributed to the strain-driven mechanism. Therefore, we have revised our manuscript and discussed the other possible factors that may contribute to the S substitution reaction such as the atomic structure of the grain boundaries, defect and dangling bonds along grain boundaries. Among them, the strain should be a major source for the high activity near the GBs. The reason is two-fold. First, there are no obvious dangling bonds at the atoms in the 4|8 dislocations, in sharp contrast to other defects, such as edges and vacancies. Second, the reaction remains energetically preferred at the growth frontier that has been progressed away from the GB (Supplementary

Fig. 16); the atoms at the frontier are so coordinated that the intrinsic chemical activity is largely attributed the lattice mismatch strain. Here we use the absorption energy of H atoms to further reflect the correlation between the chemical reactivity and strain while excluding the effect of defects. We calculate the variation of absorption energy of H atoms on a perfect MoSe₂ sheet shown in Fig. R2. The absorption energy of H atoms drops by ~0.6 eV when a moderate 3% tensile strain is applied. In comparison, the lattice strain near the GB can be up to 4% and therefore is sufficient to regulate the chemical activity thereof. We thus conclude that the chemical reactivity around the GBs is dominated by the strain. The intrinsic chemical activity of 4|8 dislocation plays a secondary role. The S substitution reaction is mainly attributed to the strain-driven mechanism as the reaction sequences also perfectly imprinted the strain patterns derived from the GBs. We have added this part in the revised manuscript and supplementary information.

Fig. R2. Evolution of absorption energy of H atoms on a MoSe₂ sheet against applied lattice strain. The adsorption energy is relative to the energy of an individual H atom. The inset presents the atomic structure (blue, grey and pink circles: Se, Mo and H atoms). The black arrow denotes the direction of strain.

Comment 2: In the caption of Fig.2c, the growth of 6-point star MoS₂ or MoSe₂?

Authors' Reply

We thank the reviewer to point out this mistake. It should be MoSe₂ and we have revised it in the manuscript.

Reviewer #4 (Remarks to the Author):

The authors report the formation of lateral heterojunction 2D TMDs by the substitution of Se with S starting from G.B.s in the matrix of MoSe₂. The manuscript is self-consistent and can be published in Nature Comm. A critical concern is about Novelty. Their method in the manuscript is not new and also not much improved in terms of quality of experimental results they've presented. The reviewer agreed with one of the reviewer's comment on Novelty. "The primary distinction in the present work is widening of initially rough boundaries into surprisingly straight sidewalls,..." In the manuscript, their analysis to support the experimental results can be considered a scientific advancement.

Authors' Reply

We highly appreciate the reviewer's positive comments on our manuscript.

For the application of lateral heterojunction as strained nanochannels, the authors choose HER and demonstrated better catalytic performance with micron sized electrochemical reactors. No objection with their experiments and results as well, but HER is not a good choice for the lateral heterojunction in 2D TMDs. It is known that surfaces and/or edge sites where dangling bonds are located show much better catalytic behaviors compared with any other in 2D TMDs. The reviewer recommend careful electrical transportation experiments through the heterojunction are much more critical and can show some potential electronic devices applications.

Authors' Reply

We thank the reviewer for the comment and suggestion. We reply them separately below:

(1) No objection with their experiments and results as well, but HER is not a good choice for the lateral heterojunction in 2D TMDs. It is known that surfaces and/or edge sites where dangling bonds are located show much better catalytic behaviors compared with any other in 2D TMDs.

Indeed, it is well accepted and proved that the dangling bonds and activity sites at GBs give rise to enhanced catalytic behavior of 2D TMDs. Nevertheless, in our HER experiments, **the key point we want**

to demonstrate is the catalytic improvement due to the precisely applied strain to GBs. In fact, prior to this work, we have demonstrated that, apart from “surfaces and/or edge sites where dangling bonds” as mentioned by the reviewer, the GBs of 2D TMDs can exhibit high catalytic activities, as shown in our recent work (Fig. R3). In that work, we demonstrate the synthesis of wafer-size atom-thin nanograin TMD films with an ultra-high-density of GBs, up to $\sim 10^{12} \text{ cm}^{-2}$. As a proof-of-concept application in electrocatalysis, devices based on our GB-rich nanograin films deliver a superior hydrogen evolution performance (onset potential: -25 mV and Tafel slope: 54 mV dec^{-1}), much better than that of the basal plane and edge devices.

Fig. R3. Engineering GBs at 2D limit for HER. **a**, Overview of the grain size and density in TMD materials obtained by various fabrication methods. **b**, TEM and STEM characterization of the MoS₂ nanograin film with a high density of GBs. **c**, The micro-electrochemical device of the MoS₂ nanograin and the corresponding polarization curves and Tafel plots for HER.

While this work elaborates on the importance of the grain boundaries for electrocatalysis, specifically, the introduction of the strain is a spontaneous process: if any GB exists in MoSe₂, the MoS₂ nano-channel can be grown precisely along this GB to introduce the strain. We compared the catalytic performance of strained GBs (“spontaneous” strained nano-channel) with un-strained GBs (single-GB in pure MoS₂), as shown in Fig. 7. The results show that the strained GBs possess better catalytic performance than that of un-strained GBs, suggesting this method is a unique strain engineering strategy to increase the activity of GBs. What’s more, the strain introduced method does not need any assistance of substrate or external force. Once the nano-channels are formed through growth, the strain is thereby applied to GBs. This is quite different from the previously reported strain engineering strategies, such as bending and stretching, which is based on the flexible substrate and external force, preventing a broad application. Considering all the above factors, we believe our HER demonstration shows some novelty in the strain engineering of TMDs.

(2) The reviewer recommend careful electrical transportation experiments through the heterojunction are much more critical and can show some potential electronic devices applications.

We fabricated three different field-effect transistors (FETs) to investigate the modification of the electrical transport properties by MoS₂ nano-channels. The configurations include pure MoSe₂ region, MoSe₂ region containing parallel (with electrodes) MoS₂ nano-channel, and MoSe₂ region containing perpendicular (to electrodes) MoS₂ nano-channel, as shown in Fig. R4a. The corresponding electrical transport transfer curves are plotted in Fig. R4b. The perpendicular device (purple curve) exhibits similar performance to the pure MoSe₂ (grey curve), suggesting that the perpendicular MoS₂ nano-channel does not affect the transportation in this direction. However, in comparison, the parallel device (orange curve) shows a slightly larger on/off ratio than that of the other two. This anisotropic transport property should be ascribed to the different electron pathways for three devices: In pure and perpendicular MoSe₂ devices, most electrons only transport in the MoSe₂ flake, and show similar performance; While in parallel device, all the electrons have to pass through the nano-channel to reach the opposite electrodes, so that the MoS₂ nano-channel with a higher bandgap than MoSe₂ lead to a larger on/off ratio. **In this consideration, the MoS₂ nano-channel can be potentially applied as the building-blocks to realize the tuning of anisotropic transport property in TMDs.**

Fig. R4. Electrical properties of MoS₂ nano-channels. **a**, Optical images of the FET devices, where electrodes parallel to the channel (upper panel) and perpendicular to the channel (lower panel) were used to study the electrical transport properties along with different directions. **b**, Electrical transportation curves of three different FETs: pure MoSe₂ device, MoSe₂ device containing parallel MoS₂ nano-channel, and MoSe₂ device containing perpendicular MoS₂ nano-channel.

Besides, to explore more potential applications, we also applied the high-resolution low-loss electron energy loss spectroscopy (LL-EELS) analysis to probe the excitons of the MoS₂ nano-channel. **We demonstrate that the lattice mismatch strain of MoS₂ nano-channel can realize the band gap engineering within 10 nm scale.** Fig. R5b shows the EELS spectra of pure monolayer MoS₂ and MoSe₂ within the absorption range from 1.4 to 2.3 eV. Two sharp peaks centered at 1.63 and 1.83 eV for MoSe₂, as well as 1.92 and 2.06 for MoS₂ should be assigned to the A and B excitons, which corresponds to the direct band gap of monolayers and the spin-orbit splitting of valence band at the K-point of the Brillouin zone, respectively¹. For the MoS₂ nano-channel, five spectra were collected at different positions across the channel, noted in Fig. R5a. As shown in Fig. R5c, at the position ~20 nm away from the nano-channel (s1 and s5), the A and B peaks of MoSe₂ can be clearly seen as expected. When the position gets closer to ~5 nm (s2 and s4), the EELS spectra still exhibit the feature of MoSe₂, except that a small broadening of the peaks can be recognized. However, when the excitons are generated in the middle of the MoS₂ nano-channel (s3), in addition to the A and B peaks of MoSe₂ due to the delocalization effect², a third peak centered at 1.99 eV appears. This peak should belong to the B excitons of MoS₂ nano-channel, and has a 70 meV redshift compared with that of pure MoS₂ (Fig. R5b), so that the A excitons of MoS₂ nano-channel overlaps with the B excitons of MoSe₂. The redshift of the excitons of MoS₂ nano-channel

arises from the strain related altering of band structure³. It is noteworthy that although the strain-induced modulation of the optical band gap of TMDs at large scale have been previously reported^{4, 5}, **our nano-channels realize the band structure engineering within 10 nm scale, offering a great potential for tunable electronic and photonic devices at ultimate length scale.** We have added this part in the revised manuscript and supplementary information.

Fig. R5. Excitonic absorption spectroscopy of MoS₂ nano-channels. **a**, ADF-STEM image of a MoS₂ nano-channel. **b**, The LL-EELS spectra of MoSe₂ and MoS₂. **c**, Five LL-EELS spectra integrated at different positions highlighted in **a**.

FET experiments. The electrodes are patterned by e-beam lithography and subsequent evaporation of Cr/Au (5/50 nm), followed by a lift-off process. The channel length is designed to be 600 nm. The devices are measured at room temperature inside a vacuum probe station using the silicon substrate as back gate. The electrical measurements are performed using Agilent B1500A semiconductor parameter analyzer with the applied drain bias of 1 V and gate bias sweeping from -50 to 50 V.

EELS experiments. EELS experiments were carried out on a Nion UltraSTEM-100. The convergence semiangle of the probe was set to 32 mrad and the EELS collection semiangle was 75 mrad. The spectra were collected with current of 2 pA, acquisition time of 500 ms and integration of 400 frame.

1. Molina-Sánchez A, Sangalli D, Hummer K, Marini A, Wirtz L. Effect of Spin-Orbit Interaction on The Optical Spectra of Single-Layer, Double-Layer, and Bulk MoS₂. *Phys. Rev. B* **88**, 045412 (2013).
2. Zhou W, Oxley MP, Lupini AR, Krivanek OL, Pennycook SJ, Idrobo J-C. Single Atom Microscopy. *Microsc. Microanal.* **18**, 1342-1354 (2012).
3. Lee J, Huang J, Sumpter BG, Yoon M. Strain-Engineered Optoelectronic Properties of 2D Transition Metal Dichalcogenide Lateral Heterostructures. *2D Mater.* **4**, 021016 (2017).
4. Conley HJ, Wang B, Ziegler JI, Haglund Jr RF, Pantelides ST, Bolotin KI. Bandgap Engineering of Strained Monolayer and Bilayer MoS₂. *Nano Lett.* **13**, 3626-3630 (2013).
5. Lloyd D, *et al.* Band Gap Engineering with Ultralarge Biaxial Strains in Suspended Monolayer MoS₂. *Nano Lett.* **16**, 5836-5841 (2016).

REVIEWERS' COMMENTS:

Reviewer #1 (Remarks to the Author):

In the revised version and response to the reviewers, the authors provide more data and understanding to demonstrate the novel potential application of the nano channel.

The authors claimed that strain induced during the structure synthesis improved the HER performance, thus the synthesis method could be important for the catalysis. However, I agree with the reviewer 4 that catalysis is not a good choice to demonstrate the uniqueness of the structure, because the desired morphology for catalysis applications is completely different from what synthesized in this work. Also, the authors mainly compared the catalytic performance of the nano channel with that of pure MoS₂. Although it is better than MoS₂, it is still not good enough for be promising in practical applications. Moreover, it is hard to tell the improved catalytic performance is due to the nano channel or defects. If it was due to the defects, there are other easier methods to introduce defects. What is the novelty of the work?

The authors performed more electrical transportation experiments. It was found that the perpendicular and parallel devices showed slightly different on/off ratio. Although they claimed this can be potentially useful for anisotropic transport property, the difference is really small which could be in the error bar range. Thus, I do not see the unique potential of the nano-channel in electronic devices.

The authors also performed EELS measurements and analysis to probe the exciton in MoS₂ nano-channel. The data is quite interesting and supports the structure characterization of the nano channel. It is hard to say how this structure will be useful in tunable electronic and photonic devices though, as the controllability and reproducibility of the synthesis has not been well addressed.

I think this work clearly showed that a new type of structures is possible during the CVD synthesis of TMDs. The characterization of the structure is solid. But the demonstrations on the applications is either over claimed or lack of evidence. From this point of view, I agree that the Novelty of this work is lacking, as the major contribution of this work to the field is limited. It is unclear what challenges this work overcame. But the data is solid and it could be published on a good journal in a specific field.

Reviewer #3 (Remarks to the Author):

The authors have addressed the concerns raised by the reviewers and thus I am now in favor of its publication.

Response to the reviewers' comments

Reviewer #1 (Remarks to the Author):

In the revised version and response to the reviewers, the authors provide more data and understanding to demonstrate the novel potential application of the nano channel.

The authors claimed that strain induced during the structure synthesis improved the HER performance, thus the synthesis method could be important for the catalysis. However, I agree with the reviewer 4 that catalysis is not a good choice to demonstrate the uniqueness of the structure, because the desired morphology for catalysis applications is completely different from what synthesized in this work. Also, the authors mainly compared the catalytic performance of the nano channel with that of pure MoS₂. Although it is better than MoS₂, it is still not good enough for be promising in practical applications. Moreover, it is hard to tell the improved catalytic performance is due to the nano channel or defects. If it was due to the defects, there are other easier methods to introduce defects. What is the novelty of the work?

The authors performed more electrical transportation experiments. It was found that the perpendicular and parallel devices showed slightly different on/off ratio. Although they claimed this can be potentially useful for anisotropic transport property, the difference is really small which could be in the error bar range. Thus, I do not see the unique potential of the nano-channel in electronic devices.

The authors also performed EELS measurements and analysis to probe the exciton in MoS₂ nano-channel. The data is quite interesting and supports the structure characterization of the nano channel. It is hard to say how this structure will be useful in tunable electronic and photonic devices though, as the controllability and reproducibility of the synthesis has not been well addressed.

I think this work clearly showed that a new type of structures is possible during the CVD synthesis of TMDs. The characterization of the structure is solid. But the demonstrations on the applications is either over claimed or lack of evidence. From this point of view, I agree that the Novelty of this work is lacking, as the major contribution of this work to the field is limited. It is unclear what challenges this work overcame. But the data is solid and it could be published on a good journal in a specific field.

Authors' Reply

We thank the reviewer for the comments. We reply them separately below:

(1) The authors claimed that strain induced during the structure synthesis improved the HER performance,

thus the synthesis method could be important for the catalysis. However, I agree with the reviewer 4 that catalysis is not a good choice to demonstrate the uniqueness of the structure, because the desired morphology for catalysis applications is completely different from what synthesized in this work. Also, the authors mainly compared the catalytic performance of the nano channel with that of pure MoS₂. Although it is better than MoS₂, it is still not good enough for be promising in practical applications. Moreover, it is hard to tell the improved catalytic performance is due to the nano channel or defects. If it was due to the defects, there are other easier methods to introduce defects. What is the novelty of the work?

a. Whether the improved catalytic performance is due to the nano-channel (induced strain)?

It should be emphasized that we have compared three sets of HER data in the manuscript: ① **the pure MoS₂**, ② **the MoS₂ GB**, and ③ **the nano-channel (Fig. 7), which correspond to perfect MoS₂ lattice, un-strained GB, and strained GB respectively**. From the Tafel plots, it can be seen that the performance of the MoS₂ GB (blue solid line in Fig. 7c) has ~ 22% improvement than that of pure MoS₂ (dark solid line in Fig. 7c), confirming that the GB can indeed improve the catalytic activity. However, more importantly, the data reveal that the performance of the nano-channel (red solid line in Fig. 7c) has additional 17% improvement than that of the MoS₂ GB (blue solid line in Fig. 7c). This improvement indicates that the strained GB (nano-channel) can exhibit even better catalytic activity than that of the un-strained GB (MoS₂ GB). **Therefore, our data clearly demonstrate the catalytic activities of these three structures: the pure MoS₂ (perfect MoS₂ lattice) < the MoS₂ GB (un-strained GB) < the nano-channel (strained GB), suggesting the strain effect in the catalytic performance improvement for HER.** Please find the summary in Table R1

	Pure MoS ₂	MoS ₂ GB	Nano-channel
Structural feature	Perfect lattice	Un-strained GB	Strained GB
Onset potential	384 mV	334 mV	266 mV
Tafel slope	139 mV dec ⁻¹	108 mV dec ⁻¹	90 mV dec ⁻¹

Table R1. Summary of the three structures and HER performance.

b. Whether the nano-channels are desired morphology for practical application?

As mentioned in previous response, our recent work (**Engineering grain boundary at 2D limit for hydrogen evolution reaction**, accepted by *Nature Communications* **11**, 57 (2020)) have already

demonstrated the synthesis of wafer-size MoS₂ films with an-ultra-high-density of GBs, which show excellent HER performance (Fig. R1). This finding offers a promisingly practical HER application of TMDs through increasing the density of GBs. **Based on this, it is believed that the strain engineering of high density GB system by the growth of nano-channels will be an efficient way to further improve the catalytic performance, so that to strengthen the possibility for potential application.**

In summary, owing to the above findings, we can conclude that undoubtedly our work possesses bold innovation and it has provided novel experimental observation that was previously not reachable.

Fig. R1. Engineering GBs at 2D limit for HER. **a**, Overview of the grain size and density in TMD materials obtained by various fabrication methods. **b**, TEM and STEM characterization of the MoS₂ nanograin film with a high density of GBs. **c**, The micro-electrochemical device of the MoS₂ nanograin and the corresponding polarization curves and Tafel plots for HER.

(2) The authors performed more electrical transportation experiments. It was found that the perpendicular and parallel devices showed slightly different on/off ratio. Although they claimed this can be potentially useful for anisotropic transport property, the difference is really small which could be in the error bar range. Thus, I do not see the unique potential of the nano-channel in electronic devices.

Thanks for the reviewer's comment. Indeed, the difference between the perpendicular and parallel devices is small. To explore more about the transport property, more sophisticated design of devices and refined measurements should be needed. However, in this manuscript, we mainly focus on the details about the growth of nano-channels and the related strain-driven mechanism. Therefore, we haven't go further into the investigation of transport property. And we will not include the transportation experiments in the manuscript.

(3) The authors also performed EELS measurements and analysis to probe the exciton in MoS₂ nano-channel. The data is quite interesting and supports the structure characterization of the nano channel. It is hard to say how this structure will be useful in tunable electronic and photonic devices though, as the controllability and reproducibility of the synthesis has not been well addressed.

Thanks for the reviewer's comment. We have removed the inappropriate description about the tunable electronic and photonic devices in the previous response, and revised the related description in the manuscript. As the reviewer's comment, the EELS measurements show some interesting results worthy of further investigation, but the EELS part is not the principal content of this work. In our future work, we will carry out in-depth study about the strain-induced band structure altering of the nano-channels, and try to explore potential applications in electronic and photonic fields.

(4) I think this work clearly showed that a new type of structures is possible during the CVD synthesis of TMDs. The characterization of the structure is solid. But the demonstrations on the applications is either over claimed or lack of evidence. From this point of view, I agree that the Novelty of this work is lacking, as the major contribution of this work to the field is limited. It is unclear what challenges this work overcame. But the data is solid and it could be published on a good journal in a specific field.

We appreciate the reviewer's positive comments on characterization and data analysis. For the application as discussed above, we believe that the HER experiments demonstrate a unique and efficient strain-related method to improve the catalytic performance of TMD GBs, making the work possess adequate novelty and

originality.

Reviewer #3 (Remarks to the Author):

The authors have addressed the concerns raised by the reviewers and thus I am now in favor of its publication.

Authors' Reply

We appreciate the reviewer's positive comments.